# Layer Freezing & Data Sieving: Missing Pieces of a Generic Framework for Sparse Training

**Geng Yuan**[1,2,†], **Yanyu Li**[1,2,†], **Sheng Li**[3], **Zhenglun Kong**[2], **Sergey Tulyakov**[1],
**Xulong Tang**[3], **Yanzhi Wang**[2], **Jian Ren**[1]
[1]Snap Inc., [2]Northeastern University, [3]University of Pittsburgh
`yuan.geng@northeastern.edu, jren@snapchat.com`

## Abstract

Recently, sparse training has emerged as a promising paradigm for efficient deep learning on edge devices. The current research mainly devotes the efforts to reducing training costs by further increasing model sparsity. However, increasing sparsity is not always ideal since it will inevitably introduce severe accuracy degradation at an extremely high sparsity level. This paper intends to explore other possible directions to effectively and efficiently reduce sparse training costs while preserving accuracy. To this end, we investigate two techniques, namely, layer freezing and data sieving. First, the layer freezing approach has shown its success in dense model training and fine-tuning, yet it has never been adopted in the sparse training domain. Nevertheless, the unique characteristics of sparse training may hinder the incorporation of layer freezing techniques. Therefore, we analyze the feasibility and potentiality of using the layer freezing technique in sparse training and find it has the potential to save considerable training costs. Second, we propose a data sieving method for dataset-efficient training, which further reduces training costs by ensuring only a partial dataset is used throughout the entire training process. We show that both techniques can be well incorporated into the sparse training algorithm to form a generic framework, which we dub SpFDE. Our extensive experiments demonstrate that SpFDE can significantly reduce training costs while preserving accuracy from three dimensions: weight sparsity, layer freezing, and dataset sieving[1].

## 1 Introduction

Sparse training, as a promising solution for efficient training on edge devices, has drawn significant attention from both the industry and academia [1]. Recent studies have proposed various sparse training algorithms with computation and memory savings to achieve training acceleration. These sparse training approaches can be divided into two main categories. The first category is fixed-mask sparse training methods, aiming to find a better sparse structure in the initial phase and keep the sparse structure constant throughout the entire training process [2, 3]. These approaches have a straightforward sparse training process but suffer from a higher accuracy degradation. Another category is Dynamic Sparse Training (DST), which usually starts the training from a randomly selected sparse structure [4, 5]. DST methods tend to continuously update the sparse structure during the sparse training process while maintaining an overall sparsity ratio for the model. Compared with the fixed-mask sparse training, the state-of-the-art DST methods have shown their superiority in accuracy and recently become a more broadly adopted sparse training paradigm [6].

---

[†]These authors contributed equally.
[1]Code will be available at https://github.com/snap-research/SpFDE.

36th Conference on Neural Information Processing Systems (NeurIPS 2022).

However, although the existing sparse training approaches can reduce meaningful training costs, most of them devote their efforts to studying how to reduce training costs by further increasing sparsity while mitigating accuracy drop. As a result, the community tends to focus on the sparse training performance at an extremely high sparsity ratio, *e.g.*, $95\%$ and $98\%$. Nevertheless, even the most recent sparse training approaches still lead to severe performance drop at these high sparsity ratios. For instance, on the CIFAR-10 dataset [7], MEST [1] has a $2.5\%$ and $4\%$ accuracy drop at $95\%$ and $98\%$ sparsity, respectively. In fact, the network performance usually begins to drop dramatically at the extremely high sparsity, while the actual gains from weight sparsity, *i.e.*, savings of computation and memory, tend to saturate. This indicates that reducing training costs by *pushing sparsity towards extreme ratios at the cost of network performance may not always the desirable methodology when a certain sparsity level has been reached.* Towards this end, we raise a fundamental question that has seldom been asked: *Are there other ways that can be seamlessly combined with sparse training to further reduce training costs effectively while maintaining network performance?*

To answer the question, we first take a step back to understand whether all layers in sparse networks are equally important. Recent studies reveal that not all the layers in *dense* Deep Neural Networks (DNNs) need to be trained equally [8, 9, 10]. Generally, the early layers in DNNs are responsible for extracting low-level features and usually have fewer parameters than the later layers. These make the early layers have higher representational similarity and converge faster during training [8, 11]. Therefore, layer freezing techniques are proposed, which stop the training (updating) of specific DNN layers early in the training process to save the training costs. The early work attempts the

Table 1: The key features of SpFDE compared to representative sparse training works.

|  | Only sparse weight | Only sparse BP | Dataset efficient training | Layer freeze |
|---|---|---|---|---|
| SNIP, GraSP | No | No | No | No |
| RigL, ITOP | Yes | No | No | No |
| SET, DSR | Yes | Yes | No | No |
| MEST | Yes | Yes | Partial | No |
| SpFDE (Ours) | Yes | Yes | Yes | Yes |

layer freezing technique in dense model training [8], while many following works focus on layer freezing in the fine-tuning process of the large transformer-based models [12, 13, 9, 10, 14, 15]. *Even with the progress, the layer freezing technique has never been explored in sparse training.*

Layer freezing seems like a promising solution for sparse training that further reduces training costs. However, the conclusion is still too early to draw, given that sparse training has two critical characteristics that make it a unique domain compared with dense DNN training and fine-tuning. This might impede the incorporation of the layer freezing technique in sparse training. ① The superiority of the DST method is attributed to its continuously changed sparse structure, which helps it end up with a better result [6]. This could inherently contradict the layer freezing that requires the layers to be unchangeable early in the training process. ② The impact of the sparsity for each layer is unknown in terms of the convergence speed. These two characteristics directly affect the feasibility and potentiality of using the layer freezing techniques in sparse training.

Back to the question we raise, whether there are other directions to effectively reduce sparse training costs besides increasing sparsity. To tackle the question, in this work, we propose two techniques that can be well incorporated with sparse training, namely layer freezing and data sieving.

- We explore the feasibility and potentiality of leveraging layer freezing in the sparse training domain by carefully analyzing the structural and representational similarity of sparse models during sparse training. We find the layer freezing technique is suitable for sparse training and has the potential to save considerable training costs (Sec. 3). Based on this, we propose a progressive layer freezing method, which is simple yet effective in saving training costs and preserving accuracy (Sec. 4.2).

- Shrinking the size of the training dataset is another possible dimension to reduce training costs. Studies have shown that the importance of each training sample is different during DNN training [16, 17]. Toneva *et al.* [18] distinguish the importance of each training sample by counting the number of times each training sample is forgotten by the network during training. Later, Yuan *et al.* [1] propose dataset efficiency in the sparse training domain with two phases. The first phase uses the whole dataset to count the forgotten times of each sample, and the second phase removes the unimportant samples. Though they prove the feasibility of dataset efficiency training in the sparse domain, it only obtains limited cost savings from its second phase. To fully exploit the potential of dataset efficiency, we propose the data sieving–a circular sieving method to dynamically update the shrunken training dataset, ensuring high dataset efficiency throughout the entire training.

Putting it all together, we propose a generic and efficient sparse training framework **SpFDE**, that achieves a significant reduction in the training computational and memory costs through *three-dimension*: weight **Sp**arsity, layer **F**reezing, and **D**ataset **E**fficiency. The comparison of key features between SpFDE and other representative sparse training works, *i.e.*, SNIP [2], GraSP [3], RigL [5], ITOP [6], SET [19], DSR [4], and MEST [1], is provided in Tab. 1. Extensive evaluation results show that the SpFDE framework consistently achieves a significant reduction in training FLOPs and memory costs while preserving a higher or similar accuracy. Specifically, SpFDE maintains the highest 71.35% accuracy at 90% sparsity and 76.03% accuracy at 80% sparsity on the CIFAR-100 and ImageNet dataset, respectively, and achieves 18% and 29% training FLOPs reduction compared to the most recent methods such as MEST [1] and ITOP [6]. Moreover, SpFDE can further save 20% ∼ 25.3% average training memory, and 42.2% ∼ 43.9% minimum required training memory compared to the prior sparse training methods.

## 2 Related Work

**Background for Sparsity Training.** Sparsity scheme and training strategy are two important components for defining a sparse training pipeline from literature.

Three main sparsity schemes introduced in the area of network pruning consist of unstructured [20, 21, 22], structured [23, 24, 25, 26, 27, 28, 29, 30, 31, 32, 33, 34, 35, 36], and fine-grained structured pruning [37, 38, 39, 40, 41, 42, 43, 44, 45, 46]. Though network pruning is initially proposed for inference acceleration, it is widely adopted in sparse training to achieve satisfactory trade-offs between network performance, *e.g.*, classification accuracy, memory footprint, and training time. Most of these works follow the training pipeline of pretraining-pruning-retraining. Instead, we consider a *generic* sparsity training framework that works for *edge devices* by focusing on sparse networks trained from scratch, instead of training dense networks, which is not feasible on resource-limited devices [1].

Research on the sparse training strategies [47, 48, 49] can be categorized into fixed-mask and sparse-mask sparse training, where the former aims to make it feasible that the training of the pruned models can be implemented on edge devices [2, 3, 50, 51, 52] and the latter studies how to reduce memory costs along with the computation during training [53, 19, 4, 54, 5]. The sparsity scheme and training strategy in this work follows MEST [1] as it does not involve dense computations, making it desirable for edge device scenarios. Unlike previous works on sparse training that mainly reduces computation by increasing the sparsity ratio at the cost of network accuracy, we investigate a new method–layer freezing–in sparse training, which can reduce training costs for an arbitrary sparsity ratio.

**Layer Freezing during Training.** The study on accelerating the training of dense neural networks has shown that not all the layers need to be trained equally and the decreasing of training iterations for certain layers can reduce training time with only minimal performance drop observed [55, 8, 12, 13, 56, 57, 9, 14, 58]. Liu *et al.* [10] and He *et al.* [15] calculate network gradients for automatically layer freezing during training. Wang *et al.* [59] use knowledge distillation [60] to guide the layer freezing schedule. However, these works require the network to be static or an extra dense network during the training, which is not eligible for the sparse training scheme. In this work, we show layer freezing can be incorporated into sparse training with detailed analysis. Additionally, we propose a flexible and hybrid layer freezing strategy that can be fitted into sparse training.

## 3 Analysis of Layer Freezing in Sparse Training

Existing works have shown the success of the layer freezing technique in *dense* model training, especially for the model fine-tuning, which effectively saves training costs in terms of computation and memory and thus accelerates the training [8, 10, 14, 15]. However, sparse training has *two critical characteristics*, *i.e.*, dynamically changed architecture and different sparsity per layer, that makes it a unique domain compared to dense DNN training and fine-tuning, which might impede incorporating the layer freezing technique into the sparse training methods. To investigate the best way to leverage layer freezing in sparse training, we need first to answer the following two questions.

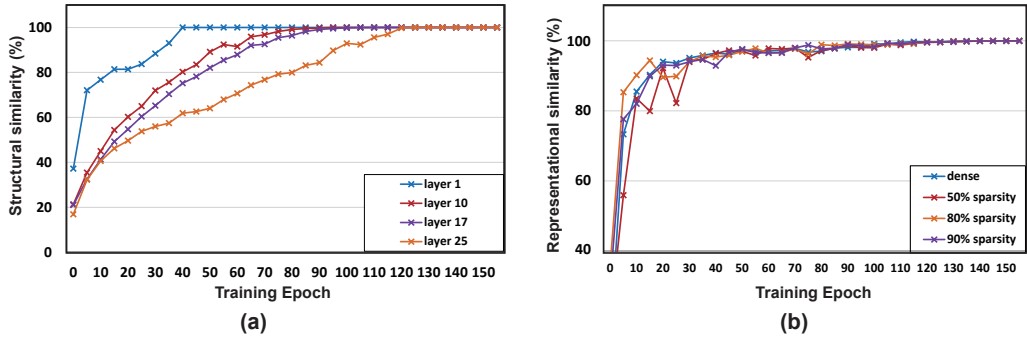

Figure 1: (a) Analysis of *structural* similarity:different layers with the same sparsity (90%). (b) Analysis of *representational* similarity: the same layer (10th) with different sparsity. All results are collected using ResNet-32 on the CIFAR-100 dataset during the sparse training process.

### 3.1 Is Layer Freezing Compatible for Dynamically Changed Network Structure?

The Dynamic Sparse Training (DST) method shows its superior performance by continuously changing its sparse model structure during training to search for a better sparse architecture, making it a desirable sparse training paradigm [5]. However, the dynamically changed network structure seems essentially *contradictory* to the layer freezing technique, given the weights of frozen layers are fixed and not further trained. Thus, these layers cannot keep searching for better sparse structures, which may compromise the quality of the sparse model trained, leading to lower model accuracy.

**Assumption.** Inspired by the convergence speed for various layers is different in conventional dense model training, we conjecture that, in DST, *the early layers may also find desired sparse structure faster than the later layers*. If true, we may be able to introduce the layer freezing technique in sparse training without compromising the sparse training accuracy.

**Experimental Setting.** We investigate the assumption by tracking the structural similarity of the sparse model along the sparse training process. Specifically, we select the well-trained sparse model as the reference model and compare the intermediate sparse model obtained at each epoch with the reference model. We define structural similarity as the percentage of the common non-zero weight locations, *i.e.*, indices, in both the intermediate sparse model and the final sparse model. For instance, the structural similarity of 70% indicates that 30% of the current intermediate sparse structure will be altered during the rest of the training process and will not be presented in the final model. The 100% structural similarity shows the sparse model structure is fully stabilized.

**Analysis Results.** Fig. 1 (a) shows the trend of structural similarity of different layers within a model along the same sparse training process. We adopt the DST method from MEST [1] with 90% unstructured sparsity and evaluate the results using ResNet-32 on the CIFAR-100 dataset. Note that we check the structural similarity by choosing the locations of the 50% most significant non-zero weights from the intermediate models. The reason is that sparse training algorithms (*e.g.*, MEST and RigL) may force less important weights/locations to be changed during sparse training, regardless of whether the sparse structure has already been stabilized. Additionally, the most significant weights play the most important role in the model's generalizability. Therefore, tracking the structural similarity using 50% most significant non-zero weights is reasonable and meaningful.

**Observation.** From the results, we can observe that the structural similarity of the first layer converges at the very early stage of the training, *i.e.*, 40 epochs, and the early layers' structural similarity converge faster than the later layers. The structural similarity of sparse training follows a similar pattern as in the dense model training. This indicates that the changing/searching of the early layers from the sparse models can be stopped earlier without compromising the quality of the final sparse model, providing the feasibility of applying the layer freezing technique in sparse training.

### 3.2 What Is the Impact of Model Sparsity on the Network Representation Learning Process?

With the above exploration giving evidence that layer freezing is compatible with sparse training (Sec. 3.1), another critical question is to understand the impact of model sparsity on the neural

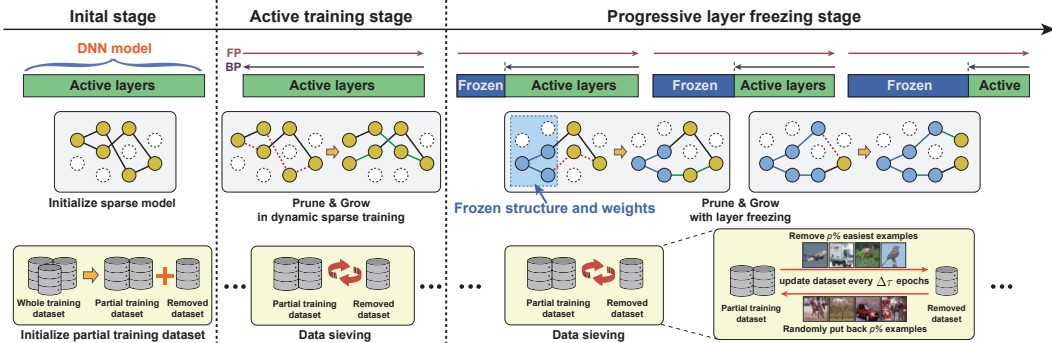

Figure 2: Overview of the SpFDE framework. The sparse training process consists of three stages. The *initial stage* creates a sparse model with a random sparse structure and randomly partitions the training dataset into a partial training dataset and a removed dataset. The *active training stage* conducts sparse training using the selected sparse training algorithm and periodically updates training dataset via data sieving. The *progressive layer freezing* stage starts to progressively freeze layers in a sequential manner and the frozen layers will not change their sparse structure and weights values.

network representation learning process. In other words, what is the convergence speed for the same layer under different sparsity? If the convergence speed is different under different sparsity, applying layer freezing would be much more complicated since deciding the stop criterion would be very challenging, and the potential gain we can have by using layer freezing could be limited.

Previous work shows that layer freezing can be used to effectively reduce training costs due to the ability of fast representation learning of the early layers in the network [8]. Another work observes that a wider network is easier to learn the representation to a saturated level [11]. Therefore, whether the sparsity would significantly slow down the representation learning or the convergence speed of the layers is unknown since the width of the layer can be changed during sparse training.

**Experimental Setting.** To explore the impact of sparsity on network representation learning speed, we adopt the centered kernel alignment (CKA) [11] as an indicator of representational similarity. We track the trend of the CKA between the final and intermediate model at each training epoch.

**Analysis Results and Observation.** The results are shown in Fig. 1 (b). We compare the CKA trends of the same layer, *i.e.*, $10^{th}$ layer in ResNet-32, in dense model training and sparse training under different sparsity. Surprisingly, we find that, in sparse training, even under a high sparsity ratio (*i.e.*, $90\%$), there is no apparent slowdown in the network representation learning speed. Similar observations can also be found in other layers (more analysis results in Appendix A). This indicates that the layer freezing technique can potentially be adopted in the sparse training process as early as in the dense model training process, thereby effectively reducing the training costs.

**Takeaway.** Considering the unique characteristics of sparse training, we first explore the feasibility and potentiality of using the layer freezing techniques in the sparse training domain. By investigating both the structural similarity and representational similarity of the sparse training, we tentatively conclude that the layer freezing technique is also suitable for sparse training and has the potential to save considerable training costs.

## 4 SpFDE Framework

In this section, we introduce our generic sparse training framework, with training costs saved through three dimensions: weight sparsity, layer freezing, and data sieving. Fig. 2 shows the overview of our SpFDE framework, and we introduce more details as follows.

### 4.1 Overview of Sparse Training Framework

The overall end-to-end training process can be logically divided into three stages, including the initial stage, the active training stage, and the progressive layer freezing stage.

**Initial Stage.** As the first stage in training, we initialize the sparse network and partial training dataset. The structure of the sparse network is randomly selected. The partial training dataset is obtained by randomly removing a given percentage of training samples from the whole training dataset, which differs from prior work that starts with a whole training dataset [1]. We ensure that only parts of the whole training dataset will be used during the entire training process.

**Active Training Stage.** Following the initial stage, all layers are actively trained (non-frozen) using a sparse training algorithm. We apply DST from MEST [1] as the sparse training method due to its superior performance, while other sparse training algorithms are compatible with our framework. We use the proposed data sieving method to update the current partial training dataset during the training periodically (more details in Sec. 4.3). Besides the computation and memory savings provided by the sparse training algorithm, our SpFDE can benefit from the data sieving method to further save computation and memory costs. Specifically, the computation costs are reduced by decreasing the number of training iterations in each epoch, and the memory costs are reduced by loading the partial dataset.

**Progressive Layer Freezing Stage.** At this stage, we begin to progressively freeze the layers in a sequential manner. The sparse structure and weight values of the frozen layers will remain unchanged during the sparse training. The computational and memory costs of all gradients of weights and gradients of activations in the frozen layers can be eliminated, which is especially crucial for resource-limited edge devices. More details are provided in Sec. 4.2.

## 4.2 Progressive Layer Freezing

Motivated by the observation that the structural and representational similarity of early layers converges faster than later layers in sparse training (Sec. 3), we propose the progressive layer freezing approach to gradually freeze layers sequentially. Specifically, a layer can be frozen only if all the layers in early of this layer are frozen. The progressive manner brings the benefits for maximizing the saving of training costs since the entire frozen part of the model does not require computing backpropagation.

### 4.2.1 Layer Freezing Algorithm

Alg. 1 shows the training flow of SpFDE and the algorithm of progressive layer freezing. For a given DNN model with $L$ layers, we divide it into $N$ blocks, with each block consisting of several consecutive DNN layers, such as a bottleneck block in the ResNet [61]. We denote $T$ as the total training epoch, $\Delta\tau$ as the sparse structure changing interval of dynamic sparse training, and $T_{frz}$ ($0 < T_{frz} < T$) as the epoch that we start the progressive layer freezing stage and freeze the first block. Then, for every $\Delta\tau$ epochs, we sequentially

---

**Algorithm 1:** Algorithm of SpFDE

**Input:** Network with randomly initialized weight $W$ under sparsity $sp$, number of blocks $N$, target training FLOPs $target\_flops$, total training epochs $T$, starting freeze epoch $T_{frz}$, and DST structure changing frequency $\Delta\tau$.

**Output:** The final sparse model.

Initialize $train\_flops$ as the total sparse training FLOPs without freezing and put all blocks in the $active\_layers$.

**for** $i = 0, \ldots, N-1$     ▷ generate freeze config
**do**
  | **if** $train\_flops > target\_flops$ **then**
  |   | $block\_list.push\_back(block_i)$
  |   | $saved\_flops =$
  |   |    BpFlops($block_i$) $* (T - T_{frz} - \Delta\tau * i)$
  |   | $train\_flops = train\_flops - saved\_flops$

**for** $epoch = 0, \ldots, T-1$     ▷ DST training loop
**do**
  | **if** ($epoch \mod \Delta\tau == 0$) **then**
  |   | **if** $epoch \geq T_{frz}$ **then**
  |   |   | $block = block\_list.pop()$
  |   |   | FreezeLayers(block)
  |   |   | $active\_layers.remove(block)$
  |   | **for** *each layer weight tensor $W^l$ in active_layers* **do**
  |   |   | $W^l \leftarrow$ Prune&Grow($W^l, sp$)
  |   | Update training dataset
  | Collect sample status

---

freeze the next block until the expected overall training FLOPs satisfy the $target\_flops$. We consider the frozen blocks still need to conduct forward propagation during training. Therefore, we compute the training FLOPs reduction of freezing a block as its sparse back-propagation computation FLOPs (calculated by BpFlops($\cdot$) in Alg. 1) multiplied by the total frozen epochs of the block. For detailed implementation, given the target training FLOPs $target\_flops$ and the total number of layers to freeze, we empirically choose to freeze 2/3 layers of the model, and the $T_{frz}$ can be easily calculated.

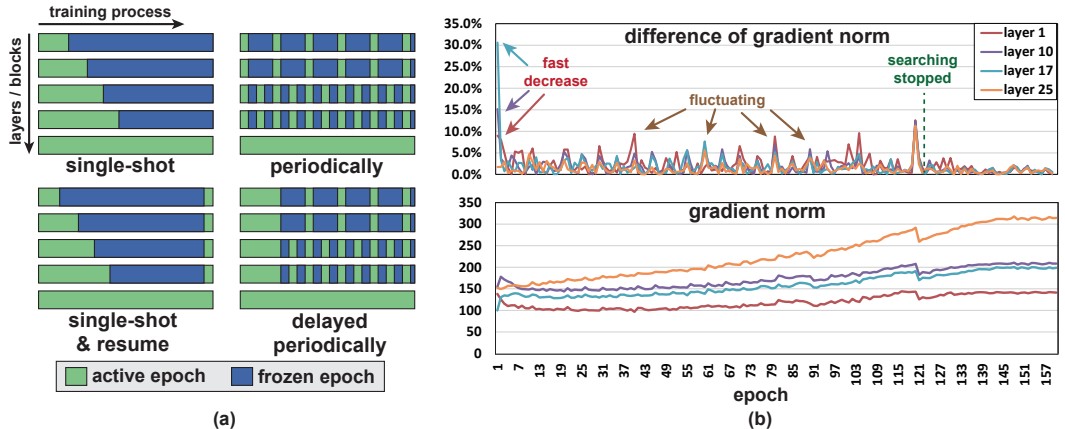

Figure 3: (a) Illustration for different layer freezing schemes. (b) The trend of layer gradient norm and the difference of layer gradient norm during dynamic sparse training.

To better combine with the DST and make sure the layers/blocks are appropriately trained before being frozen, we synchronize the progressive layer freezing interval to the structure changing interval, *i.e.*, $\Delta\tau$, of the sparse training, and adopt a layer/block-wise cosine learning rate schedule according to the total active training epoch of each layer/block.

### 4.2.2 Design Principles for Layer Freezing

There are two key principles for deriving a layer freezing algorithm, the freezing scheme and the freezing criterion. Here we discuss the reasons that the proposed progressive layer freezing is rational.

**Freezing Scheme**. Since sparse training may target the resource-limited edge devices, it is desired to have the training method as simple as possible to reduce the system overhead and strictly meet the budget requirements. Therefore, we follow a cost-saving-oriented freezing principle to guarantee the target training costs and derive the layer freezing scheme, which can include the *single-shot*, *single-shot & resume*, *periodically freezing*, and *delayed periodically freezing* (as illustrated in Fig. 3 (a)). We adopt the *single-shot* scheme since it achieves the highest accuracy under the same training FLOPs saving (detailed results in Appendix B). The possible reason is that the single-shot freezing scheme has the longest active training epochs at the beginning of the training, which helps layers converge to a better sparse structure before freezing.

**Freezing Criterion**. With the freezing scheme decided, another important question is how to derive the freezing criterion, *i.e.*, choosing which iterations or epochs to freeze the layers. Existing works have explored adaptive freezing methods by calculating and collecting the gradients during the fine-tuning of dense networks [10]. However, from our observations, the unique property of sparse training makes these approaches not applicable. For example, as shown in Fig. 3 (b), the difference of gradients norms from different layers decreases very fast at the beginning of the sparse training while it keeps fluctuating after some epochs because of the prune-and-grow weights. Abstracting the freezing criterion based on the gradient norm would inevitably introduce extra computation and system complexity since the changing patterns of the gradient norm difference are volatile. Therefore, our strategy of combining the layer freezing interval with the DST interval is more favorable.

### 4.3 Circular Data Sieving

We propose the data-sieving method to achieve true dataset-efficient training throughout the sparse training process. As shown in Fig. 2, at the beginning of the training, we randomly remove a given portion of total training samples (e.g., 20%) from the training dataset depending on the target training cost reduction ratio and create a partial training dataset and a removed dataset. During the sparse training, for every $\Delta\tau$ epoch, we update the current partial training dataset by removing the easiest $p\%$ of the training sample from the partial training dataset and adding them to the removed dataset. Then, we retrieve the same number of samples from the removed dataset and add them back to the partial training dataset to keep the total number of training samples unchanged.

Table 2: Comparison of classification accuracy (on CIFAR-100) and training FLOPs ($\times e^{15}$) between the proposed SpFDE and the most representative sparse training works using ResNet-32.

| Method \ Sparsity | 90% | | 95% | | 98% | |
|---|---|---|---|---|---|---|
| | FLOPs ($\downarrow$) | Acc. ($\uparrow$) | FLOPs ($\downarrow$) | Acc. ($\uparrow$) | FLOPs ($\downarrow$) | Acc. ($\uparrow$) |
| LTH [62] | N/A | 68.99 | N/A | 65.02 | N/A | 57.37 |
| SNIP [2] | 1.32 | 68.89 | 0.66 | 65.22 | 0.26 | 54.81 |
| GraSP [3] | 1.32 | 69.24 | 0.66 | 66.50 | 0.26 | 58.43 |
| DeepR [53] | 1.32 | 66.78 | 0.66 | 63.90 | 0.26 | 58.47 |
| SET [19] | 1.32 | 69.66 | 0.66 | 67.41 | 0.26 | 62.25 |
| DSR [4] | 1.32 | 69.63 | 0.66 | 68.20 | 0.26 | 61.24 |
| MEST [1] | 1.54 | 71.30 | 0.96 | 70.36 | 0.38 | 67.16 |
| SpFDE$_{15\%+15\%}$ | 1.26 | 71.35±0.28 | 0.66 | 70.43±0.22 | 0.30 | 67.04±0.20 |
| SpFDE$_{20\%+20\%}$ | 1.12 | 71.25±0.23 | 0.58 | 70.14±0.17 | 0.26 | 66.37±0.24 |
| SpFDE$_{25\%+25\%}$ | 0.96 | 71.02±0.39 | 0.52 | 69.48±0.19 | 0.24 | 65.04±0.19 |

We adopt the number of forgetting times as the criteria to indicate the complexity of each training sample. Specifically, for each training sample, we collect the number of forgetting times by counting the number of transitions from being correctly classified to being misclassified within each $\Delta\tau$ interval. We recollect this number for each interval to ensure the newly added samples can be treated equally. Additionally, we use the structure changing frequency $\Delta\tau$ in sparse training as the dataset update frequency to minimize the impact of the changed structure on the forgetting times.

We treat the removed dataset as a queue structure, retrieving samples from its head and adding the newly removed sample to its tail. After all the initial removed samples are retrieved, we shuffle the removed dataset after each update, making all the training samples can be used at least once. As a result, we can gradually sieve the relatively easier samples out and only use the important samples for dataset-efficient training. More analysis results are in the Appendix C.

## 5 Experimental Results

In this section, we evaluate our proposed SpFDE framework on benchmark datasets, including CIFAR-100 [7] and ImageNet [63], for the image classification task with ResNet-32 and ResNet-50. Note that we follow the previous work [1, 3, 2] using the $2\times$ widen version of ResNet-32. We compare the accuracy, training FLOPs, and memory costs of our framework with the most representative sparse training works [2, 3, 53, 54, 4, 19, 5, 1] at different sparsity ratios. Models are trained by using PyTorch [64] on an $8\times$A100 GPU server. We adopt the standard data augmentation and the momentum SGD optimizer. Layer-wise cosine annealing learning rate schedule is used according to the frozen epochs. To make a fair comparison with the reference works, we also use 160 training epochs on the CIFAR-100 dataset and 150 training epochs on the ImageNet dataset. We choose MEST+EM&S [1] as our training algorithm for weight sparsity since it does not involve any dense computations, making it desirable for edge device scenarios. We apply uniform unstructured sparsity across all the convolutional layers while only keeping the first layer dense. More experiments on other datasets and the detailed hyper-parameter setting are provided in the Appendix D.

### 5.1 Comparison on Model Accuracy and Training FLOPs

Tab. 2 shows the comparison of accuracy and computation FLOPs results on CIFAR-100 dataset using ResNet-32. Each accuracy result is averaged over 3 runs. We denote the configuration of our SpFDE using $x\% + y\%$, where $x$ indicates the target training FLOPs reduction during layer freezing and $y$ is the percentage of removed training data. Our SpFDE can consistently achieve higher or similar accuracy compared to the most recent sparse training methods while considerably reducing the training FLOPs. Specifically, at 90% sparsity ratio, SpFDE$_{20\%+20\%}$ maintains similar accuracy as MEST [1], while achieving 27% training FLOPs reduction. When compared with DeepR [53], SET [19], and DSR [4], SpFDE$_{25\%+25\%}$ achieves 27% FLOPs reduction and $+1.36\% \sim +4.24\%$ higher accuracy. More importantly, when comparing SpFDE$_{25\%+25\%}$ at 90% sparsity with the MEST at 95% sparsity, we can see the two methods have the same training FLOPs, *i.e.*, 0.96, while SpFDE$_{25\%+25\%}$ has a clear higher accuracy, *i.e.*, $+0.66\%$. This further strengthens our motivation

Table 3: Comparison results on ImageNet using ResNet-50 with unstructured sparsity scheme.

| Method | Training FLOPs ($\times$e18) | Inference FLOPs ($\times$e9) | Top-1 Accuracy |
|---|---|---|---|
| Dense | 3.2 | 8.2 | 76.9 |
| Sparsity ratio | | 80% | |
| SNIP [2] | 0.74 | 1.7 | 72.0 |
| GraSP [3] | 0.74 | 1.7 | 72.1 |
| DeepR [53] | n/a | n/a | 71.7 |
| SNFS [54] | n/a | n/a | 74.2 |
| DSR [4] | 1.28 | 3.3 | 73.3 |
| SET [19] | 0.74 | 1.7 | 72.9 |
| RigL [5] | 0.74 | 1.7 | 74.6 |
| SpFDE$_{20\%+20\%}$ | **0.74** | 1.7 | **75.35** |
| MEST [1] | 1.17 | 1.7 | 75.70 |
| SpFDE$_{15\%+15\%}$ | **0.84** | 1.7 | **75.75** |
| RigL-ITOP [6] | 1.34 | 1.7 | 75.84 |
| SpFDE$_{10\%+10\%}$ | **0.95** | 1.7 | **76.03** |

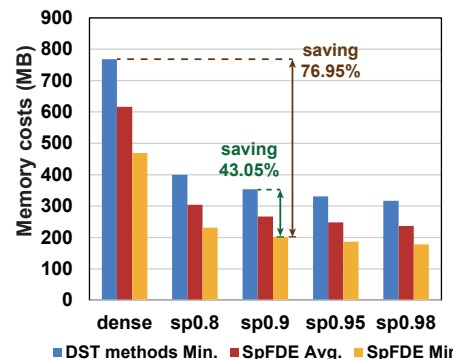

Figure 4: The comparison of the training memory costs between the minimum required memory of DST methods and the average and minimum costs of SpFDE under the configuration of saving 20% training FLOPs via layer freezing.

that *pushing sparsity towards extreme ratios is not the only desirable direction for reducing training costs.* SpFDE provides new dimensions to reduce training costs while preserving accuracy.

Tab. 3 provides the comparison results on the ImageNet dataset using ResNet-50. At each training FLOPs level, SpFDE consistently achieves higher accuracy than existing works. It is interesting to see that SpFDE outperforms the original MEST, in both accuracy and FLOPs saving. The FLOPs saving is attributed to layer freezing and data sieving for end-to-end dataset-efficient training. Moreover, compared to the one-time dataset shrinking used in MEST, our data sieving dynamically updates the training dataset, mitigating over-fitting and resulting in higher accuracy. We also conduct ablation studies on the impact of only applying layer freezing technique or data sieving technique and the results are provided in Appendix E.

## 5.2 Reduction on Memory Cost

From Fig. 4, we can see the superior memory saving of our SpFDE framework. The memory costs indicate the memory footprint used during the sparse training process, including the weights, activations, and the gradient of weights and activations, using a 32-bit floating-point representation with a batch size of 64 on ResNet-32 using CIFAR-100. The "SpFDE Min." stands for the training memory costs after all the target layers are frozen, while the "SpFDE Avg." is the average memory costs throughout the entire training process. The baseline results of "DST methods Min." only consider the minimum memory costs requirement for DST methods [2, 3, 53, 54, 4, 19, 5, 1, 6], which ignores the memory overhead such as the periodic dense back-propagation in RigL [5], dense sparse structure searching at initialization in [2, 3], and the soft memory bound in MEST [1]. Even under this condition, our "SpFDE Avg." can still outperform the "DST methods Min." with a large margin ($20\% \sim 25.3\%$). The "SpFDE Min." results show our minimum memory costs can be reduced by $42.2\% \sim 43.9\%$ compared to the "DST methods Min." at different sparsity ratios. This significant reduction in memory costs is especially crucial to edge training.

## 5.3 Discussion and Limitation

The reduction in training FLOPs of our method comes from three sources: weight sparsity, frozen layers, and shrunken dataset. However, the actual training acceleration depends on different factors, *e.g.*, the support of the sparse computation, layer type and size, and system overhead. Generally, the same FLOPs reduction from the frozen layers and shrunken dataset can lead to higher actual training acceleration than weight sparsity (more details in Appendix F). This makes our layer freezing and data sieving method more valuable in sparse training. We use the overall computation FLOPs to measure the training acceleration, which may be considered a theoretical upper bound.

## 6 Conclusion

This work investigates the layer freezing and data sieving technique in the sparse training domain. Based on the analysis of the feasibility and potentiality of using the layer freezing technique in sparse training, we introduce a progressive layer freezing method. Then, we propose a data sieving technique, which ensures end-to-end dataset-efficient training. We seamlessly incorporate layer freezing and data sieving methods into the sparse training algorithm to form a generic framework named SpFDE. Our extensive experiments demonstrate that our SpFDE consistently outperforms the prior arts and can significantly reduce training FLOPs and memory costs while preserving high accuracy. While this work mainly focuses on the classification task, a future direction is to further investigate the performance of our methods on other tasks and networks. Another exciting topic is studying the best trade-off between these techniques when considering accuracy, FLOPs, memory costs, and actual acceleration.

## Acknowledgment

This work is supported in part by National Science Foundation CNS-1909172.

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
