# Appendix

## A  More Results for the Impact of Model Sparsity on the Network Representation Learning Process

Sec. 3.2 of the main paper discusses the impact of model sparsity on the network representation learning process. Here we provide more experimental results. Specifically, we evaluate the representational similarity using the CKA value of the same layer from the model (ResNet-32) with different sparsity at each epoch and compare them with the final model. We choose the early (1st and 3rd), middle (18th), and late layers (25th and 32nd) to track their CKA trends. We evaluate three sparsity ratios, including medium (50%), medium-high (80%), and high (90%) sparsity. The results are shown in Fig. A1.

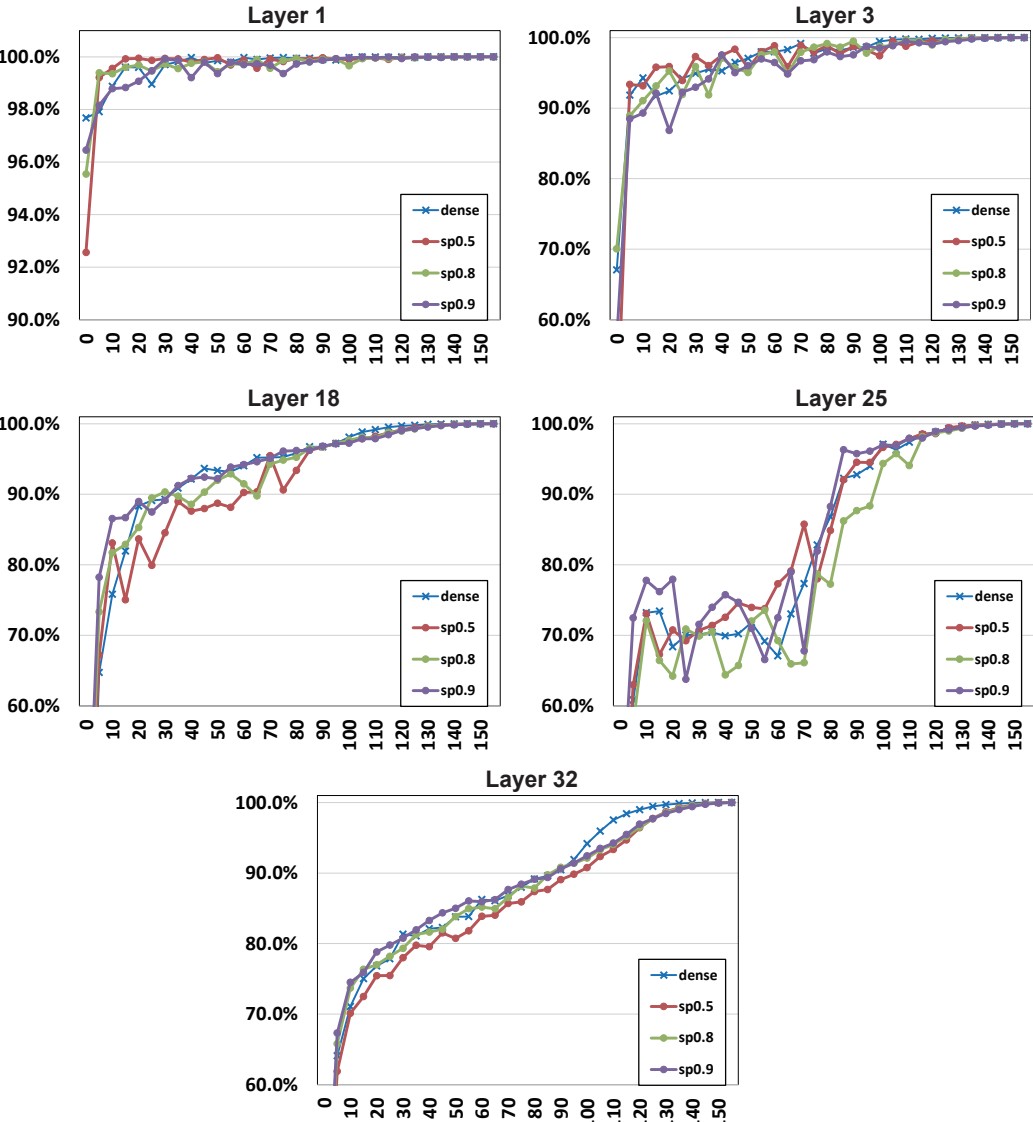

Figure A1: Analysis of *representational* similarity: the same layer (1st, 3rd, 18th, 25th, and 32nd) with different sparsity at different epochs. All results are collected using ResNet-32 on the CIFAR-100 dataset during the sparse training process.

We can observe that the representation learning speed of sparse training under different sparsity ratios is similar to the dense model training at each layer. This indicates that layer sparsity does not slow

down the layer representation learning speed. Therefore, the layer freezing technique can potentially achieve considerable training FLOPs reduction in sparse training domains similar to the dense model training.

# B   Ablation Analysis on Freezing Schemes

In our work, we evaluate four different types of freezing schemes (Sec. 4.4.4 of the main paper), including the *single-shot freezing*, *single-shot freezing & resume*, *periodically freezing*, and *delayed periodically freezing*.

**Single-Shot Freezing Scheme.** The single-shot freezing scheme is the default freezing scheme used in our progressive layer freezing method. For this scheme, we progressively freeze the layers/blocks in a sequential manner, as shown in Alg. 1 and Fig. 3 (a) in the main paper.

**Single-Shot Freezing & Resume Scheme.** This scheme follows the same way to decide the per layer/block freezing epoch as the single-shot scheme, except that we make the freezing epoch for all layers/blocks $t$ epochs earlier and resume (defrost) the training for all layers/blocks at the last $t$ epochs. In this case, we can keep the single-shot & resume has the same FLOPs reduction as the single-shot scheme, and the entire network can be fine-tuned at the end of training with a small learning rate.

**Periodically Freezing Scheme.** For the periodically freezing scheme, we let the selected layers freeze periodically with a given frequency so that all the layers/blocks are able to be updated at different stages of the training process. The basic idea is to let the early layers/blocks updated (trained) less frequently than the later layers. For example, we let the early layers/blocks only be updated for one epoch in every four epochs and let the middle layers/blocks only be updated for one epoch in every two epochs. Therefore, we consider the update frequency of the early and middle layers/blocks are $1/4$ and $1/2$, respectively. To ensure that when a layer is frozen, all the layers in early of it are frozen, we need to let the freezing period be the numbers of power of two (*e.g.*, 2, 4, and 8). In our experiments, we divide the ResNet32 into three blocks and set the update frequency for the first and second blocks to $1/4$ and $1/2$, respectively. The last block will not be frozen during the training. We control the number of layers in each block to satisfy the overall training FLOPs reduction requirement.

**Delayed Periodically Freezing Scheme.** For this scheme, we first let all the layers/blocks be trained actively for certain epochs, then periodically freeze the layers used in the periodically freezing scheme. To achieve the same training FLOPs reduction as the periodically freezing scheme, more layers are needed to be included in the first and second blocks.

Table A1: Comparison of classification accuracy between different freezing schemes using ResNet32 on the CIFAR-100 dataset. The $20\%$ FLOPs reduction in the table is only attributed to the layer freezing and does not count the weight sparsity.

| Sparsity | 60% | | 90% | |
|---|---|---|---|---|
| Freeze Scheme | FLOPs Reduction | Accuracy | FLOPs Reduction | Accuracy |
| Non-Freeze | - | 73.68±0.43 | - | 71.28±0.34 |
| Single-Shot | 20% | **73.61**±0.19 | 20% | **71.30**±0.17 |
| Single-Shot & Resume | 20% | 73.49±0.26 | 20% | 71.21±0.21 |
| Periodically | 20% | 72.78±0.23 | 20% | 70.86±0.44 |
| Delayed Periodically | 20% | 72.88±0.13 | 20% | 70.95±0.43 |

**Accuracy Comparison.** Tab. A1 shows the accuracy comparison of different freezing schemes at medium ($60\%$) and high ($90\%$) sparsity ratio. We use the ResNet32 on the CIFAR-100 dataset. Our target training FLOPs reduction through layer freezing is set to $20\%$. We do not use the data sieving technique in this experiment. The results show that the single-shot scheme consistently achieves the highest accuracy at both $60\%$ and $90\%$ sparsity ratio. The accuracy of the single-shot freezing & resume scheme is slightly lower than the single-shot scheme and the two periodically freezing schemes are the worst. These demonstrate that the layer freezing technique in sparse training prefers

to train the layers/blocks as good as possible at the beginning of the training, and the "last-minute" or periodic fine-tuning does not benefit the final accuracy.

## C  Data Sieving Analysis

### C.1  Basic Concepts of Dataset Efficient Training

We use the number of forgetting events [18, 1] as the criteria to measure the difficulty of the training examples. A forgetting event can be defined as a training sample that goes from being correctly classified to being misclassified by a network in two consecutive training epochs. The training examples that have a higher number of forgetting events throughout the training indicate the examples are more complex and are considered more informative to the training. On the contrary, the training examples that have a lower number of forgetting events or have never been forgotten are relatively easier examples and are less informative to the training. Removing the unforgettable examples from the training dataset does not harm the training accuracy [18].

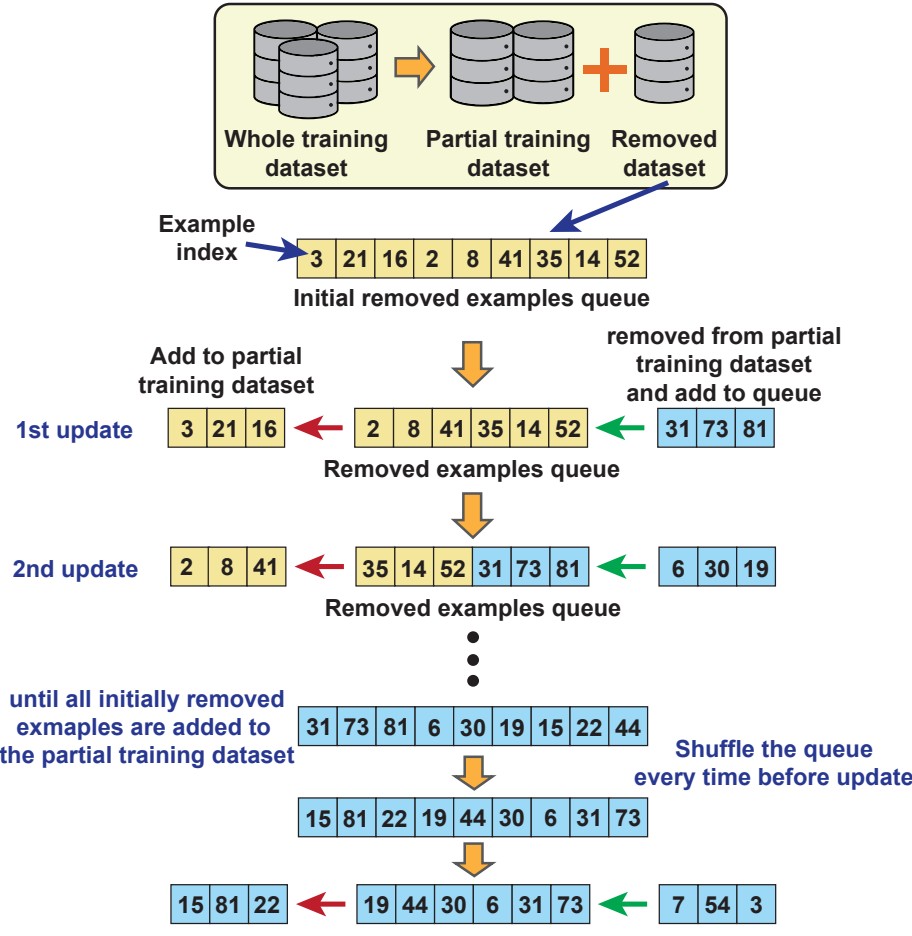

Figure A2: Data sieving process.

## C.2 More Details about the Proposed Data Sieving Method

Fig. A2 shows the detailed update process of our data sieving method. We use a queue data structure to contain the indices of the removed examples. For each time the partial training dataset is updated, we retrieve the examples from the head of the removed examples queue and add the newly removed examples to the tail of the removed examples queue. To ensure all the examples are at least added to the partial training dataset once, we do not shuffle the queue until all the initial removed examples are added back to the partial training dataset.

Table A2: Accuracy comparison of SpFDE under different data sieving update ratios. The update ratio is the percentage of the number of examples in the removed dataset. Results are obtained using ResNet32 on the CIFAR-100 dataset.

| Update ratio | 10% | 20% | 30% | 50% |
|---|---|---|---|---|
| remove 15% | 70.68 | 71.12 | 71.35 | 70.92 |
| remove 20% | 70.69 | 71.03 | 71.25 | 70.86 |
| remove 25% | 70.42 | 70.99 | 71.02 | 70.63 |

Tab. A2 shows an ablation study on the data sieving update ratio. The number of updated examples in each dataset update process is proportional to the number of examples in the removed dataset. The update ratio in the table denotes the percentage of examples retrieved from the removed dataset and added to the partial training dataset. We evaluate different update ratios (*i.e.*, 10%, 20%, 30%, and 50%) under different dataset removal ratios (15%, 20%, and 25%). From the results, we can find that a 30% update ratio is the most desirable setting for the data sieving, which achieves the highest accuracy under different dataset removal ratios.

Table A3: Hyper-parameter settings.

| Experiments | CIFAR-10/100 | | ImageNet | |
|---|---|---|---|---|
| Basic training hyper-parameter settings | | | | |
| Training epochs ($\tau_{end}$) | 160 | | 150 | |
| Batch size | 32 | | 1024 | |
| Learning rate scheduler | cosine | | cosine | |
| Initial learning rate | 0.15 | | 1.024 | |
| Ending learning rate | 4e-8 | | 0 | |
| Momentum | 0.9 | | 0.875 | |
| $\ell_2$ regularization | 1e-4 | | 3.05e-5 | |
| Warmup epochs | 0 | | 8 | |
| DST-related (MEST [1]) hyper-parameter settings | | | | |
| Num of epochs do structure search | 120 | | 120 | |
| Structure change frequency ($\Delta\tau$) | 5 | | 2 | |
| Prune&Grow schedule with target final sparsity s | 0 - 90: | GR (s - 0.05) RM (s) | 0 - 90: | GR (s - 0.05) RM (s) |
| | 90 - 120: | GR (s - 0.025) | 90 - 120: | GR (s - 0.025) |
| PruneTo sparsity (RM) | | RM (s) | | RM (s) |
| GrowTo sparsity (GR) | 120 - 160: | No search | 120 - 150: | No search |

Table A4: The epoch $T_{frz}$ that starts the progressive layer freezing stage for different target training FLOPs reduction for ResNet32 on CIFAR-10/100.

| Target FLOPs saving | 10% | 15% | 20% | 25% |
|---|---|---|---|---|
| $T_{frz}$ | 80 | 70 | 60 | 60 |

Table A5: The epoch $T_{frz}$ that starts the progressive layer freezing stage for different target training FLOPs reduction for ResNet50 on ImageNet.

| Target FLOPs saving | 7.5% | 10% | 15% | 20% | 22% |
|---|---|---|---|---|---|
| $T_{frz}$ | 90 | 80 | 60 | 50 | 50 |

## D   Hyper-Parameter and More Experimental Results

**Detailed Experiment Setup.** Tab. A3 shows detailed hyper-parameters regarding the general training and dynamic sparse training. In our work, we use the MEST-EM&S [1] as our base sparse training algorithm. To make fair comparisons to the reference works, we also use the $2\times$ widened version ResNet-32 in our work, which is the same as all the baseline works shown in Tab. 2 and Tab. A6. In our data sieving method, we remove the easiest $p\%$ training examples from the partial training dataset every time we update our training dataset. In our experiments, we make the $p\%$ equals to the 30% of the number of examples in the removed dataset. Tab. A4 and Tab. A5 show the epoch $T_{frz}$ that starts the progressive layer freezing stage for different target training FLOPs reduction for ResNet32 on CIFAR-10/100 and ResNet50 on ImageNet, respectively.

**More Results on the CIFAR-10 Dataset.** Tab. A6 shows the accuracy comparison of our SpFDE and the most representative sparse training works using ResNet32 on the CIFAR-10 dataset. Our SpFDE consistently achieves higher or similar accuracy on the CIFAR-10 dataset compared to the most recent sparse training methods while considerably reducing the training FLOPs.

**More Results on the ImageNet Dataset.** Tab. A7 shows the accuracy comparison using ResNet50 on the ImageNet dataset at the $90\%$ sparsity ratio. At the similar training FLOPs level ($0.32 \sim 0.36 \times 10^{18}$), our SpFDE achieves $73.81\%$ on top-1 accuracy, outperforming the best baseline work MEST by $1.45\%$.

Table A6: Comparison of classification accuracy and training FLOPs ($\times e^{15}$) between the proposed SpFDE and the most representative sparse training works using ResNet-32 on CIFAR-10 dataset.

| Method \ Sparsity | 90% | | 95% | | 98% | |
|---|---|---|---|---|---|---|
| | FLOPs ($\downarrow$) | Acc. ($\uparrow$) | FLOPs ($\downarrow$) | Acc. ($\uparrow$) | FLOPs ($\downarrow$) | Acc. ($\uparrow$) |
| LTH [62] | N/A | 92.31 | N/A | 91.06 | N/A | 88.78 |
| SNIP [2] | 1.32 | 92.59 | 0.66 | 91.01 | 0.26 | 87.51 |
| GraSP [3] | 1.32 | 92.38 | 0.66 | 91.39 | 0.26 | 88.81 |
| DeepR [53] | 1.32 | 91.62 | 0.66 | 89.84 | 0.26 | 86.45 |
| SET [19] | 1.32 | 92.3 | 0.66 | 90.76 | 0.26 | 88.29 |
| DSR [4] | 1.32 | 92.97 | 0.66 | 91.61 | 0.26 | 88.46 |
| MEST [1] | 1.54 | 93.27 | 0.96 | 92.44 | 0.38 | 90.51 |
| SpFDE$_{10\%+10\%}$ | 1.42 | 93.24±0.22 | 0.88 | 92.45±0.27 | 0.35 | 90.33±0.30 |
| SpFDE$_{15\%+15\%}$ | 1.26 | 92.99±0.26 | 0.66 | 92.21±0.29 | 0.30 | 89.67±0.16 |
| SpFDE$_{20\%+20\%}$ | 1.12 | 92.50±0.08 | 0.58 | 91.82±0.17 | 0.26 | 89.51±0.14 |

Table A7: Accuracy comparison using ResNet-50 on ImageNet at 90% sparsity.

| Method | Training FLOPs ($\times$e18) | Inference FLOPs ($\times$e9) | Top-1 Accuracy |
|---|---|---|---|
| Dense | 3.2 | 8.2 | 76.9 |
| Sparsity ratio | | 90% | |
| SNIP [2] | 0.32 | 0.82 | 67.2 |
| GraSP [3] | 0.32 | 0.82 | 68.1 |
| DeepR [53] | n/a | n/a | 70.2 |
| SNFS [54] | n/a | n/a | 72.3 |
| DSR [4] | 0.96 | 2.46 | 71.6 |
| SET [19] | 0.32 | 0.82 | 70.4 |
| RigL [5] | 0.32 | 0.82 | 72.0 |
| RigL-ITOP [6] | 0.8 | 0.82 | 73.8 |
| MEST$_{0.5\times}$ | 0.37 | 0.82 | 72.36 |
| SpFDE$_{22\%+22\%}$ | **0.36** | 0.82 | **73.81** |
| SpFDE$_{15\%+15\%}$ | **0.47** | 0.82 | **74.40** |
| SpFDE$_{10\%+10\%}$ | **0.52** | 0.82 | **74.93** |
| MEST [1] | 0.60 | 0.82 | 75.1 |
| SpFDE$_{7.5\%+7.5\%}$ | **0.55** | 0.82 | **75.14** |

# E  Ablation Study on Layer Freezing and Data Sieving

We also conduct ablation studies for the impact of layer-freezing and data sieving on accuracy by themselves (Tab. A8 and Tab. A9). The results are obtained using ResNet-32 on the CIFAR-100 with the sparsity of 60% and 90%. The accuracy results are the average value of 3 runs using random seeds.

Table A8: Ablation analysis on different **layer freezing** ratios. The accuracy results are obtained using ResNet-32 on the CIFAR-100 with the sparsity of 60% and 90%, respectively.

| FLOPs reduction | None | 10% | 15% | 20% | 25% | 27.5% | 30% | 32.5% | 35% |
|---|---|---|---|---|---|---|---|---|---|
| sparsity 60% | 73.97 | 74.05 | 74.09 | 73.76 | 73.27 | 73.14 | 73.03 | 72.36 | 72.00 |
| sparsity 90% | 71.30 | 71.33 | 71.31 | 71.29 | 71.18 | 71.08 | 70.82 | 70.35 | 70.26 |

Table A9: Ablation analysis on different **data sieving** ratios. The accuracy results are obtained using ResNet-32 on the CIFAR-100 with the sparsity of 60% and 90%, respectively.

| FLOPs reduction | None | 10% | 15% | 20% | 25% | 27.5% | 30% | 32.5% | 35% |
|---|---|---|---|---|---|---|---|---|---|
| sparsity 60% | 73.97 | 73.98 | 73.94 | 73.88 | 73.66 | 73.68 | 73.58 | 73.55 | 73.20 |
| sparsity 90% | 71.3 | 71.36 | 71.30 | 71.33 | 71.11 | 71.09 | 70.98 | 70.86 | 70.59 |

From the experiments, we can further find some interesting observations:

- Under both sparsity of 60% and 90%, saving up to 15% training costs (FLOPs) via either layer freezing or data sieving does not lead to any accuracy drop.

- When under a higher sparsity ratio (90% vs. 60%), sparse training can tolerate a higher FLOPs reduction for both layer freezing and data sieving. For example, compared to the non-freezing case (i.e., None in the second column), the layer freezing with a 20% FLOPs reduction leads to a -0.01% and -0.21% accuracy drop for 90% sparsity and 60% sparsity, respectively. As for the data sieving, compared to the non-freezing case, under a 20% FLOPs reduction, there is a -0.19% and -0.31% accuracy drop for 90% sparsity and 60% sparsity, respectively. The possiable reason is that, under a higher sparsity ratio, the upper bound

for model accuracy/generalization capability is decreased, mitigating the sensitivity to the number of training data or layer freezing.

- With a relatively higher FLOPs reduction ratio (i.e., 30% 35%), data sieving preserves higher accuracy than layer freezing under the same FLOPs reduction ratio. This inspires that if people intend to pursue a more aggressive FLOPs reduction at the cost of accuracy degradation, removing more data via the data sieving method is a more desirable choice than freezing more layers.

Furthermore, in Tab. A10, we show a comparison between only using layer-freezing or data sieving, or both of them to achieve similar FLOPs reductions.

Table A10: Analysis of **layer freeze**, **data sieving**, or **both of them** for similar FLOPs reduction. The accuracy results are obtained using ResNet-32 on the CIFAR-100.

|  | Freeze + Data Sieve | Freeze only | Data Sieve only |
|---|---|---|---|
| FLOPs reduction | 27.75% (15%+15%) | 27.50% | 27.50% |
| Accuracy | 71.35 | 71.08 | 71.09 |
| FLOPs reduction | 36% (20%+20%) | 35.00% | 35.00% |
| Accuracy | 71.25 | 70.26 | 70.59 |

It can be observed that to achieve similar FLOPs reduction, using layer-freezing and data sieving together achieves much higher accuracy than by only using one of them individually, showing the importance of combining the two techniques.

Table A11: Training acceleration analysis on layer freezing by using ResNet32 on CIFAR-100.

| FLOPs reduction (Layer freezing) | baseline | 10% | 15% | 20% | 25% |
|---|---|---|---|---|---|
| Epoch time (s) | 46.94 | 42.75 | 40.53 | 38.10 | 35.83 |
| Acceleration | - | 8.93% | 13.66% | 18.83% | 23.67% |

Table A12: Training acceleration analysis on data sieving by using ResNet32 on CIFAR-100.

| FLOPs reduction (Partial dataset) | baseline | 10% | 15% | 20% | 25% |
|---|---|---|---|---|---|
| Epoch time (s) | 46.94 | 42.65 | 40.19 | 37.98 | 35.49 |
| Acceleration | - | 9.14% | 14.38% | 19.09% | 24.39% |

# F   Discussion on Acceleration

In our work, the reduction in training FLOPs comes from three sources: weight sparsity, frozen layers, and shrunken dataset. It is well-known that the acceleration based on weight sparsity is heavily affected by many different factors, such as the sparse computation support from a sparse matrix multiplication library or the dedicated compiler optimizations [39]. Besides, the sparsity schemes play an important role in the sparse computation acceleration. Currently, the actual acceleration by leveraging weight sparsity is still limited even at a very high sparsity ratio [1].

We also evaluate the acceleration achieved by using our layer freezing and data sieving methods. We measure the training time over 50 consecutive training epochs for each configuration and calculate the average value.

Tab. A11 and Tab. A12 show the acceleration results by using our layer freezing and data sieving methods, respectively. We compare the per epoch training latency with different FLOPs saving configurations (*i.e.*, 10%, 15%, 20%, and 25%) with the baseline result (*i.e.*, using whole dataset and without freezing). We can see that both methods achieve almost the linear training acceleration

according to the FLOPs reduction. This indicates that both methods only introduce negligible overhead to the training process. Compared to the weight sparsity, this demonstrates the superiority of layer freezing and data sieving methods in the acceleration efficiency when under the same FLOPs reduction. Most importantly, the layer freezing and data sieving methods have a high degree of practicality since the acceleration can be easily achieved using native PyTorch/TensorFlow without additional support.