# OpenReview forum: "Layer Freezing & Data Sieving: Missing Pieces of a Generic Framework for Sparse Training"
_NeurIPS.cc/2022/Conference — NeurIPS 2022 Accept_

### Official Review · Reviewer_AURL · 2022-07-10

**Rating:** 3
**Confidence:** 5
**Soundness:** 1 poor
**Presentation:** 2 fair
**Contribution:** 2 fair

**Summary:**

The authors propose to modify an existing dynamic sparse training (DST) method, MEST, by (a) freezing the training of early layers in a CNN  sequentially early in training to remove the requirements to calculate/maintain backwards pass, and (b) use a “data sieving” method to train in a more data efficient manner. The freezing method is proposed under the intuition that early layers converge to both weights and sparse structure earlier than later layers, and the authors present experimental results claiming to demonstrate this. The authors compare their method (SpFDE) to existing sparse training methods, including SNIP, GrasP, SET and MEST on CIFAR-100 with ResNet-32.

**Questions:**

Questions:
* The experiment presented in 3.1/fig 1(a) comparing the masks of two models trained with a DST method seems to assume that the mask found by DST methods is the same for a DNN trained with different initializations - this is well known not to be the case (as also seen with lottery tickets)! How is your experiment finding masks that are substantially similar, up to 100% in some cases? Are the initializations the same? What is the saliency measure used when choosing the “50% most significant weights”?
* The experiment presented in 3.2/fig 1(b) claims that sparsity levels do not affect sparse training convergence time with MEST. Is this a known result? At least with other DST methods, such as RiGL, I believe this may not be the case.
How exactly do you calculate the theoretical FLOPSin general, and what are your assumptions on the FLOPS of a frozen layer? On a sparse layer?
* You state that you focus on sparse training from scratch rather than pre-training as it’s not feasible to use pre-training on edge devices. Why? Intuition suggests the exact opposite if anything.

Suggestions:
* I actually find the research direction(s) presented by this paper to be very interesting and intuitive, especially the freezing direction, however the current paper doesn’t have sufficient experimental analysis, results or focus to make for a convincing story. I would encourage the authors to take the feedback of the reviewers and attempt to correct this for a future submission.
* I would recommend focusing on each of these methods as it’s own paper (or only on the freezing), they are not related enough to write together as one method, and it weakens the story and clarity of your research motivation and findings. There are significant experimental details missing likely as a result.
* Your work is not the first work to explore “frozen” layers (e.g. fine-tuning, or freezing weights) & sparsity, I would remove this claim as it’s hard to justify. Also, FreezeNet in particular should be cited here as other work looking at freezing weights in pruning/sparsity.
* “front layers” -> “early layers” in all instances, to match the common terminology in the field.

**Limitations:**

The authors do not discuss any limitations or potential impacts of their work. I would note specifically that the data sieving method could affect the learning of biases from datasets since it is sub-sampling the dataset during training, and the authors do not identify this.

**Strengths And Weaknesses:**

Strengths:
* Questions the wisdom of using the same amount of data/training time to train different layers in a sparse DNN training given the common intuition that early layers converge on weights faster.
* Interesting and intuitive hypothesis that the early layers in a CNN may converge on sparse structure early in training with a dynamic sparse training method.
* Analysis of working memory footprint of model is interesting, and rare to find in papers.

Weaknesses:
* Only theoretical FLOPS shown, and the calculations of theoretical FLOPS for frozen layers is flawed in that it doesn’t consider the real-world scenarios of usage of these models. Forward passes are still required for training through the frozen layers, and this analysis seems to ignore this. Theoretical FLOPS are at best misleading and often completely unrelated to real-world efficiency (e.g. unstructured sparse masks of 99% have a theoretical FLOPS that is low, but hard to accelerate), and any paper claiming to demonstrate real-world speed up of training/inference must include real-world computational measurements of timings. Unfortunately, this fact alone means this paper cannot be accepted as-is. It is not explained what FLOPs these are (e.g. multiply-accumulate), or how they are calculated in general, or specifically for frozen layers.
* Results showing decreased FLOPS with good generalization only demonstrated on a single model (ResNet-32) and on a single dataset (CIFAR-100). Results on CIFAR-10 and ImageNet are in the appendix, but seem to show increased FLOPS for similar/better generalization.
* Missing substantial experimental details, making some of the experiments non-reproducible. In fact, some of the results of experiments presented contradict prior work/well-known issues in sparse training (see questions).
* The first few layers in a CNN are actually upwards of 90% of the compute of the whole model (while being a small fraction of the weights as the authors ). In this sense, the intuition presented by the authors on why they focus on the latter layers doesn’t make sense.
* By trying to present two very different approaches to improving sparse training as one method, the paper reads as a confusing story — these two methods are not related enough to roll into one paper/method IMHO.
* SpFDE incorporates both the frozen layers and data sieving methods, and results are only shown as such. To evaluate the individual contribution of each of these methods the authors need to provide an ablation analysis, however there is none. It is impossible to evaluate the claims of the authors unfortunately on each of the individual proposed methods without this. Furthermore, the results comparing the proposed method with other DST and sparse training methods are not directly comparable, since they are not using data sieving.
* Experiments only focus on one specific dynamic sparse training method (MEST), but conclusions are made over all dynamic sparse training methods.

---

> ### Author Response · Authors · 2022-08-02
> **Author Response to Reviewer AURL (Part 6/6)**
>
> **Q7. Applying data sieving to more dynamic sparse training algorithms.**
>
> Thank you for the kind suggestion. Our layer-freezing and data sieving methods are also compatible with other dynamic sparse training algorithms. Here are the results of adopting layer-freezing and data sieving in RigL.
>
> >**Table R. Accuracy results of adopting layer-freezing and data sieving in RigL. The results are obtained using ResNet32 on the CIFAR-100 dataset.**
> | FLOPs reduction (freeze + data sieve) |  None | 10%+10% | 15%+15% | 20%+20% | 25%+25% |
> |--------------------------------------:|------:|--------:|--------:|--------:|--------:|
> |               FLOPs reduction (total) |  None |     19% |     28% |     36% |     44% |
> |                          sparsity 90% | 70.45 |   70.53 |   70.50 |   70.56 |   69.96 |
> |                          sparsity 95% | 69.52 |   69.67 |   69.42 |   68.20 |   67.71 |
> |                          sparsity 98% |  63.2 |   63.31 |   63.26 |   62.54 |   61.70 |
>
> It can be observed that our method can further reduce 28% training FLOPs on top of sparsity while having no or minor accuracy degradation. Specifically, under 90% sparsity, our method can even save extra 36% training FLOPs without accuracy degradation.
> These results show that our method can also work well on the other dynamic sparse training method.
> We also want to kindly mention that this training FLOPs reduction is independent of the model sparsity and **can achieve nearly linear real-world acceleration according to the FLOPs reduction.**
>
> ---
>
> **Q8. You state that you focus on sparse training from scratch rather than pre-training as it’s not feasible to use pre-training on edge devices. Why? Intuition suggests the exact opposite if anything.**
>
> In our paper, we mention that  *“Most of these works follow the training pipeline of pretraining-pruning-retraining. Instead, we consider a generic sparsity training framework that works for edge devices by focusing on sparse networks trained from scratch, instead of the pretraining of dense networks, which is not feasible on resource-limited devices.”*
>
> We intend to mention that we do not consider training dense models via dense training on edge devices, which is not friendly for resource-limited devices.
>
> We can use the pretrained dense model and sparsify it as the initial sparse model for sparse training, which may provide higher accuracy. However, in our paper, we follow the standard paradigm of the sparse training works that train a model from scratch, so that we can have fair comparisons with the other works.

---

> > ### Comment · Reviewer_AURL · 2022-08-09
> > **Rebuttal Response**
> >
> > (7) These are very interesting as this was my main concern with the experimental results, e.g. in table 3.
> >
> > First, I think given your previous rebuttal it's very clear this statement is hard to back? "... and can achieve nearly linear real-world acceleration according to the FLOPs reduction..." real-world and FLOPS are not the same thing as we agreed, so I would be careful with this claim, as it's misleading.
> >
> > I think this is a *much* better story for your work and results, i.e. not comparing your work to methods like RiGL as if they are competing methods, but comparing existing methods with/without data sieving and/or layer freezing. This would be a very substantial change to your existing paper however, and not something to be done in your rebuttal as a minor edit unfortunately.
> >
> > (8) I'm still confused on this... any training on edge is very difficult, and it's not clear to me why fine-tuning a model would be worse than training a model from scratch (even if it's sparse), so I would personally just remove this statement as I believe it's not backed up by any evidence or actual experiments you did.

---

> > > ### Author Response · Authors · 2022-08-09
> > > **Author response**
> > >
> > > (7) First, our understanding is that the reviewer thinks a better story for our paper is that we should conduct a huge amount of experiments to show the proposed techniques can be used by any dynamic sparse training methods. However, we humbly think the current organization of our paper is **more suitable** than what the reviewer suggested for the following reasons.
> > >
> > > - We mainly conduct experiments on MEST since it is the **state-of-the-art** dynamic sparse training algorithm. We think it is important to show that layer freezing and data sieving can work well using the state-of-the-art dynamic sparse training algorithm.
> > >
> > > - We humbly think it is not necessary to validate our method on all the sparse training algorithms, since the performance of many previous works is not as good as the most recent ones. We **don’t have to** run the experiments to show our proposed method is compatible **with every old method** since we didn’t claim our methods could work on any sparse training algorithms.
> > >
> > > - Showing our method works well on RigL can be an extra experiment to support that our proposed method is **robust and general**. Our method has the potential to be applied to a wide range of dynamic sparse training algorithms.
> > >
> > > - Demonstrating the compatibility of our method with other previous methods can help us prove the generalizability and robustness of our method. However, it **should be better considered an ablation study rather than the main results of the paper**.
> > >
> > > - The main results should show the best results to **push the cutting edge** of the domain. We believe this is more **important and informative to the readers**.
> > >
> > > ---
> > >
> > > Second, our claim is not misleading. The "FLOPs reduction (total)" in Table R is the FLOPs reduction achieved by layer-freezing + data sieving, not including the FLOPs reduction by sparsity. Our statement "We also want to kindly mention that this training FLOPs reduction is independent of the model sparsity and can achieve nearly linear real-world acceleration according to the FLOPs reduction." is appropriate since we do not take the FLOPs reduction of sparsity into account. We have experimental results (Table A8&9) to support this which is agreed by you.
> > >
> > > ---
> > >
> > > (8)
> > > We have never said that fine-tuning a model is worse than training from scratch. We do not understand why the reviewer is confused about this.
> > > Here we would like to provide some common knowledge of sparse training. The general motivation of all sparse training works is based on the condition that training or fine-tuning a dense model via a **dense training process** is not friendly to the resource-limited devices. The reason is that dense training requires storing dense weights and gradients, leading to huge memory costs. Dense training also requires dense computations, leading to high energy consumption. On the contrary, sparse training can be a promising solution to solve these problems.
> > > We follow the standard sparse training paradigm that trains sparse models from scratch (using randomly initialized models). Of course, we can also use a pretrained model as our initial model, then use a sparse training algorithm for sparse fine-tuning.

---

> ### Author Response · Authors · 2022-08-02
> **Author Response to Reviewer AURL (Part 5/6)**
>
> **Q6. Ablation for layer freezing and data sieving.**
>
> Thank you for the valuable suggestion. Here we provide the ablation analysis for the impact of layer-freezing and data sieving on accuracy by themselves (Table O and Table P). The results are obtained using ResNet32 on the CIFAR100 dataset. We perform the evaluation on the sparsity of 60% and 90%. The accuracy results are the average value of 3 runs using different random seeds.
>
> >**Table O. Ablation analysis on different layer freezing ratios**
> |  FLOPs reduction | None  | 10%   | 15%   | 20%   | 25%   | 27.5% | 30%   | 32.5% | 35%   |
> |--------------|-------|-------|-------|-------|-------|-------|-------|-------|-------|
> | sparsity 60% | 73.97 | 74.05 | 74.09 | 73.76 | 73.27 | 73.14 | 73.03 | 72.36 | 72.00 |
> | sparsity 90% | 71.30 | 71.33 | 71.31 | 71.29 | 71.18 | 71.08 | 70.82 | 70.35 | 70.26 |
>
> >**Table P. Ablation analysis on different data sieving ratios**
> | FLOPs reduction | None  | 10%   | 15%   | 20%   | 25%   | 27.5% | 30%   | 32.5% | 35%   |
> |--------------|-------|-------|-------|-------|-------|-------|-------|-------|-------|
> | sparsity 60% | 73.97 | 73.98 | 73.94 | 73.88 | 73.66 | 73.68 | 73.58 | 73.55 | 73.20 |
> | sparsity 90% |  71.3 | 71.36 | 71.30 | 71.33 | 71.11 | 71.09 | 70.98 | 70.86 | 70.59 |
>
>
> From the experiments, we can further find some interesting observations:
> 1. Under both sparsity of 60% and 90%, saving up to 15% training costs (FLOPs) via either layer freezing or data sieving does not lead to any accuracy drop.
>
> 2. When under a higher sparsity ratio (90% vs. 60%), sparse training can tolerate a higher FLOPs reduction for both layer freezing and data sieving. For example, compared to the non-freezing case (i.e., None in the second column), the layer freezing with a 20% FLOPs reduction leads to a -0.01% and -0.21% accuracy drop for 90% sparsity and 60% sparsity, respectively. As for the data sieving, compared to the non-freezing case, under a 20% FLOPs reduction, there is a -0.19% and -0.31% accuracy drop for 90% sparsity and 60% sparsity, respectively. We think the reason is, under a higher sparsity ratio, the upper bound for model accuracy/generalization capability is decreased, mitigating the sensitivity to the number of training data or layer freezing.
>
> 3. With a relatively higher FLOPs reduction ratio (i.e., 30%~35%), data sieving preserves higher accuracy than layer freezing under the same FLOPs reduction ratio. This inspires that if people intend to pursue a more aggressive FLOPs reduction at the cost of accuracy degradation, removing more data via the data sieving method is a more desirable choice than freezing more layers.
>
>
> We further show a comparison between only using layer-freezing or data sieving, or both of them to achieve similar FLOPs reductions, in the following table (Table Q).
>
>
> >**Table Q. Analysis of layer freeze, data sieving, or both of them for FLOPs reduction**
> |  | freeze + data sieve | freeze only | data sieve only |
> |---:|---:|---:|---:|
> | FLOPs reduction | 27.75% (15%+15%) | 27.50% | 27.50% |
> | Accuracy  | 71.35 | 71.08 | 71.09 |
> | FLOPs reduction | 36% (20%+20%) | 35.00% | 35.00% |
> | Accuracy  | 71.25 | 70.26 | 70.59 |
>
>
> It can be observed that to achieve similar FLOPs reduction, **using layer-freezing and data sieving together achieves much higher accuracy than by only using one of them individually**, showing the importance of combining the two techniques.
>
> We will definitely include these ablation studies and these interesting observations in our final version. We hope our work can motivate more future research in the sparse training area.

---

> > ### Comment · Reviewer_AURL · 2022-08-09
> > **Rebuttal Response**
> >
> > (6) Thanks to the authors for these results, I appreciate running new experiments is never easy.
> >
> > For me, contrary to the authors, these results bring more questions on why these two methods are being used as one they do not achieve appear to be highly complementary, or result in "much higher accuracy". In Table Q, on CIFAR100 a 0.27 percentage point increase in accuracy may not be significant (need to evaluate over multiple random inits), and even if it is, it's extremely small.
> >
> > Table O and P, I'm going to restrict my analysis to 90% sparsity since that's simply the only sparsity level at which acceleration would ever practically make sense. Data sieving seems to be the main winner between these two methods when looked at independently, and of course this could be applied to any sparse training method as far as I'm aware. Layer freezing does seem to have a significant impact on generalization for moderate FLOPS decrease.
> >
> > Unfortunately with these results I'm more concerned that it doesn't make sense, for example in Table 3, to be comparing SpFDE with other sparse training methods when the main win seems to be from data sieving alone.

---

> > > ### Author Response · Authors · 2022-08-09
> > > **Author response**
> > >
> > > First, we want to clarify that all the accuracy results in the paper and rebuttal are averaged over **three training processes with different random seeds**.
> > >
> > > Second, regarding the reviewer’s statement that “*In Table Q, on CIFAR100 a 0.27% percentage point increase in accuracy may not be significant*”, we respectfully think it is inappropriate. The accuracy gap between the three methods (i.e., (1) both freezing + data sieving, (2) freezing only, (3) data sieving only) is becoming more and more significant as the FLOPs saving ratio increases. Therefore, **it is important to use both layer freezing and data sieving to achieve high FLOPs reduction, instead of using one of them.**
> > >
> > > The reviewer questioned the reason for combining layer freezing and data sieving based on the claim that the 0.27% accuracy gap under the 27.75% FLOPs reduction is not significant. However, such a claim is not accurate since the reviewer **ignores the fact** that combining the two methods achieves a **more significant accuracy advantage** as the FLOPs reduction ratio goes higher. For example, from the results in Table Q with 36% FLOPs reduction, we can see that combining freezing and data sieving achieves 0.66%~0.99% higher accuracy than only using one of them.
> > >
> > > Third, we respectfully disagree with the reviewer that data sieving is the main winner. By looking at Tables O and P, we can see that when the FLOPs reduction is less or equal to 27.5% with 90% sparsity, layer freezing and data sieving have **very similar results**. For example, under 25% FLOPs reduction with 90% sparsity ratio, layer freezing has higher accuracy than data sieving, i.e., 71.18% vs. 71.11%. Therefore, the conclusion from the reviewer that data sieving seems to be the main winner is not correct. Additionally, in Table 3, we show the experiments with 10%+10%, 15%+15%, and 20%+20% FLOPs reduction from layer freezing and data sieving, where the two techniques contribute similarly to the computation saving and model performance under such FLOPs reduction ratio, as can be seen in Table O and P. Therefore, we humbly think the claim from the reviewer that data sieving contributes alone to the results in Table 3 is not accurate.

---

> ### Author Response · Authors · 2022-08-02
> **Author Response to Reviewer AURL (Part 4/6)**
>
> **Q4. Some of the results of experiments presented contradict prior work/well-known issues in sparse training**
>
> >**Q4.2. Sec 3.1/fig 1(b) Representational Similarity: Sparsity levels do not affect representation learning speed in MEST. Is this a known result? What about RigL?**
>
> To the best of our knowledge, there is no prior literature providing solid conclusions on this.
> This is the reason that we conduct experiments and investigate by ourselves.
>
> As suggested, we also conduct the same experiments using RigL instead of MEST.
> Similar observations can be obtained. That is, the sparsity ratio will not lead to an apparent change in the network representation learning speed, as shown in the following Table M.
>
> >**Table M. Trend of representational similarity: the same layer (10th) with different sparsity. RigL is used for dynamic sparse training.**
> | sparsity \ epoch | 1 | 20 | 40 | 60 | 80 | 100 | 120 | 140 | 160 |
> |---|---:|---:|---:|---:|---:|---:|---:|---:|---:|
> | Dense | 0.150 | 0.903 | 0.958 | 0.969 | 0.968 | 0.981 | 0.998 | 0.999 | 1.000 |
> | sp0.5 | 0.200 | 0.834 | 0.941 | 0.958 | 0.969 | 0.990 | 0.992 | 0.999 | 1.000 |
> | sp0.8 | 0.151 | 0.899 | 0.924 | 0.972 | 0.988 | 0.988 | 0.994 | 0.997 | 1.000 |
> | sp0.9 | 0.251 | 0.908 | 0.966 | 0.971 | 0.977 | 0.981 | 0.992 | 0.999 | 1.000 |
>
> ---
>
> **Q5. The first few layers in a CNN are actually upwards of 90% of the compute of the whole model (while being a small fraction of the weights as the authors ). In this sense, the intuition presented by the authors on why they focus on the latter layers doesn’t make sense.**
>
> Regarding the comment that “first few layers in a CNN are actually upwards of 90% of the compute of the whole model. … focus on the latter layers doesn’t make sense.”, we would like to kindly clarify that:
> 1. The situation that the first few layers take up most of the computation costs has become less common in recent CNNs. The computation costs are quite evenly distributed throughout the model. We provide two examples of ResNet50 and MobileNetV2 that use the input size of 224x224 (ImageNet) in Table N:
>
> >**Table N. The percentage of a layer’s computation costs compared to the entire model.**
> | model \ layer No. | 1 | 2 | 3 | 4 | 21 | 22 | 23 | 24 | 41 | 42 | 43 | 44 |
> |---|---:|---:|---:|---:|---:|---:|---:|---:|---:|---:|---:|---:|
> | ResNet50 | 0.6% | 0.6% | 0.6% | 2.5% | 2.5% | 2.5% | 0.6% | 2.5% | 2.5% | 0.6% | 2.5% | 5.1% |
> | MobileNetV2 | 3.6% | 1.3% | 2.0% | 6.2% | 0.7% | 1.6% | 0.3% | 1.6% | 0.1% | 1.6% | 2.6% | 0.3% |
>
> 2. We freeze the layers in a sequential manner (from earlier layers to the later layers) because the earlier layers converge faster than the later layers. By nature, the earlier layers are responsible for low-level feature extraction, which become well-trained earlier.

---

> > ### Comment · Reviewer_AURL · 2022-08-09
> > **Rebuttal Response**
> >
> > (4b) Interesting results! Unfortunately a bit out of context in your work's motivation, but I think this is a promising research direction as I would have assumed the learning speed was related to sparsity.
> >
> > (5)
> >
> > (i) Unfortunately I'm quite certain this analysis is incorrect for ResNet50 at least... I've done this analysis myself for ResNet 50. Just intuitively your analysis should seem off though, you are doing convolutions over a very large image size compared to the rest of the model which operates on much smaller spatial-feature maps. This makes me worry about how you are computing FLOPS here and/or your model, please explain how you calculated this.
> >
> > (ii) This is a better way of motivating this, I would use only this motivation in your paper instead.

---

> > > ### Author Response · Authors · 2022-08-09
> > > **Author response**
> > >
> > > (4b) We respectfully disagree with the reviewer that the exploration of representational similarity “is out of context in our work’s motivation”.
> > > In our paper, we explore both the trends of structural similarity and representational similarity of sparse training. As we clearly mentioned in our paper, the different convergence speeds of structural similarity demonstrate the **feasibility** of using the layer freezing technique in sparse training.
> > >
> > > The exploration of representational similarity in sparse training shows the layers under different sparsity ratios have similar representation learning speeds, indicating we do not need to delay the freezing time compared to the dense model training. This demonstrates the **potentiality** of the layer freezing technique in sparse training with different sparsity ratios to save considerable training costs.
> > >
> > > **Once again, the exploration of structural similarity and representational similarity shows the feasibility and potentiality of layer-freezing in sparse training. This is perfectly aligned with our motivation, and both Reviewer y6sL and Reviewer EhKj agreed that our work is nicely motivated.**
> > >
> > > ---
> > >
> > > (5)
> > > (i)
> > > The input image/featuremap size is reduced from 224x224 to 56x56 (by using 7x7 CONV stride=2, and pooling) after the first CONV and pooling layer, so the earlier layers do not have that heavy computation costs.
> > >
> > > The FLOPs of a layer is computed by 2 * (kernel_size * kernel_size) * input_channel * output_feature_size * output_feature_size * output_channel.
> > >
> > > Specifically, we follow the standard way to calculate the FLOPs, which consider the multiplication and accumulation as 2 floating-point operations, that is the reason we have “2 *” in the equation.
> > >
> > > **The FLOPs results in the paper are correct.**
> > >
> > > **Could you please let us know what the results are on your side if you have different numbers?**
> > >
> > > ---
> > >
> > > (ii)
> > > We humbly think the reviewer still misunderstood our work. The motivation of our work is that we aim to explore other directions to save training costs for sparse training besides increasing the sparsity ratio.
> > >
> > > With the motivation bear in mind, we propose two main techniques, layer freezing and data sieving. Each technique has its own implementation details. The earlier layers are responsible for low-level feature extraction is **only the motivation for a specific step of layer freezing**, i.e., freezing the layers sequentially. This can only motivate us to freeze the layers sequentially when we have decided to use layer freezing. However, it is not the motivation for our work.
> > >
> > > The motivation of a specific step in layer freezing is not the motivation of our work. We hope the reviewer can understand the differences between them.

---

> ### Author Response · Authors · 2022-08-02
> **Author Response to Reviewer AURL (Part 3/6)**
>
> **Q4. Some of the results contradict prior work in sparse training**
>
> >**Q4.1. Sec 3.1/fig 1(a) seems to assume that the mask found by DST is the same for a DNN trained with different initializations. Masks that are similar up to 100%. Are the initializations the same? Why choose the “50% most significant weights”?**
>
> We respectfully think the reviewer might have a misunderstanding of Figure 1(a) that leads to the confusion.
>
> We fully agree that sparse training using different initialization will converge to different sparse masks. However, for results shown in Figure 1(a), we did not use different initialization to train two models.
>
> As mentioned in Lines 150-157,  *“We select the well-trained sparse model as the reference model and compare the intermediate sparse model obtained at each epoch with the reference model.”* We only focus on one **single** sparse training process. For example, if the training takes 160 epochs. Then, we use the final model (at epoch 160) as the reference model (denoted as model_ep160). We compare the intermediate model at each epoch during **the same** training process  (model_ep1, …, model_ep159) to the reference model (model_ep160).
>
> So, we can track the sparse masks convergence during training process. Therefore, it is reasonable that the structural similarity will always converge to 100% at the end. Also, the faster convergence speed of the earlier layers indicates that they tend to find desirable sparse masks faster and remain stable earlier in DST. This provides us the feasibility to freeze the layers without compromising the accuracy.
>
> The reason that we use 50% most significant weights is that the sparse training algorithm, i.e., MEST, forces the 50% less important weights/mask to be changed, and the most significant weights play the most important role in the model. Thus, tracking the structural similarity using 50% most significant weights is reasonable.
>
> We provide the analysis of structural similarity traces measured using 70% most significant non-zero weights and all (100%) non-zero weights in Table I and Table J. All the results are based on 90% weight sparsity.
>
> >**Table I. Structure similarity (%) using 70% most significant non-zero weights (with MEST)**
> | layer \ epoch |    1 |   20 |   40 |   60 |   80 |   100 |   120 |   140 |   160 |
> |---------------|-----:|-----:|-----:|-----:|-----:|------:|------:|------:|------:|
> | layer 1       | 37.5 | 73.4 | 89.2 | 93.8 | 97.5 | 100.0 | 100.0 | 100.0 | 100.0 |
> | layer 10      | 20.8 | 48.2 | 67.6 | 84.7 | 87.6 |  93.5 | 100.0 | 100.0 | 100.0 |
> | layer 17      | 19.9 | 46.2 | 63.4 | 78.3 | 83.7 |  91.8 |  98.2 | 100.0 | 100.0 |
> | layer 25      | 15.3 | 42.5 | 51.5 | 63.0 | 71.0 |  82.7 |  95.6 | 100.0 | 100.0 |
>
>
> >**Table J. Structure similarity (%) using 100% non-zero weights (with MEST)**
> | layer \ epoch |    1 |   20 |   40 |   60 |   80 |  100 |   120 |   140 |   160 |
> |---------------|-----:|-----:|-----:|-----:|-----:|-----:|------:|------:|------:|
> | layer 1       | 30.2 | 65.1 | 76.7 | 82.6 | 91.3 | 94.8 | 100.0 | 100.0 | 100.0 |
> | layer 10      | 19.5 | 40.2 | 56.7 | 73.6 | 85.7 | 89.3 |  94.5 | 100.0 | 100.0 |
> | layer 17      | 19.7 | 39.1 | 52.7 | 66.0 | 78.6 | 82.1 |  90.3 | 100.0 | 100.0 |
> | layer 25      | 13.6 | 37.4 | 46.5 | 54.1 | 63.6 | 73.1 |  81.3 |  89.2 | 100.0 |
>
> From the two tables, the earlier layers converge significantly faster than the later layers. Therefore, the conclusion from using 70% and 100% most significant non-zero weights is consistent with using 50% of the most significant non-zero weights.
>
>
> Additionally, we evaluate the structural similarity using RigL in Table K and L (90% sparsity):
>
> >**Table K. Structure similarity (%) using 50% non-zero weights (with RigL)**
> | layer \ epoch |    1 |   20 |   40 |    60 |    80 |   100 |   120 |   140 |   160 |
> |---------------|-----:|-----:|-----:|------:|------:|------:|------:|------:|------:|
> | layer 1       | 39.5 | 93.0 | 97.7 | 100.0 | 100.0 | 100.0 | 100.0 | 100.0 | 100.0 |
> | layer 10      | 37.7 | 77.8 | 89.5 |  94.4 |  99.3 | 100.0 | 100.0 | 100.0 | 100.0 |
> | layer 17      | 20.7 | 65.8 | 76.2 |  90.3 |  97.4 | 100.0 | 100.0 | 100.0 | 100.0 |
> | layer 25      | 19.1 | 60.4 | 65.1 |  89.3 |  92.1 | 100.0 | 100.0 | 100.0 | 100.0 |
>
> >**Table L. Structure similarity (%) using 70% non-zero weights (with RigL)**
> | layer \ epoch |    1 |   20 |   40 |   60 |    80 |   100 |   120 |   140 |   160 |
> |---------------|-----:|-----:|-----:|-----:|------:|------:|------:|------:|------:|
> | layer 1       | 31.7 | 86.7 | 93.3 | 98.3 | 100.0 | 100.0 | 100.0 | 100.0 | 100.0 |
> | layer 10      | 31.8 | 63.5 | 73.5 | 84.3 |  92.5 |  98.8 | 100.0 | 100.0 | 100.0 |
> | layer 17      | 20.2 | 50.6 | 63.4 | 79.5 |  87.1 |  97.9 | 100.0 | 100.0 | 100.0 |
> | layer 25      | 17.7 | 49.1 | 64.0 | 74.2 |  85.4 |  97.5 | 100.0 | 100.0 | 100.0 |
>
> We will modify the paper accordingly to further clarify the experimental setting of Figure 1 (a) in the final version.

---

> > ### Comment · Reviewer_AURL · 2022-08-09
> > **Rebuttal Response**
> >
> > (4) Indeed I did not understand the same initialization was being used here. Even reading the highlighted passage in the paper that you say explains this, I do not understand this to be the case still, and unfortunately I don't think most readers would. The fact the same initialization is used (and specifically what type of initialization this is) should be explicitly stated in the description of the experiment given the importance of it in the context of sparse training. This was not the only issue I had in the lack of experimental details however, and I believe this still needs a lot more work to be reproducible.

---

> > > ### Author Response · Authors · 2022-08-09
> > > **Author response**
> > >
> > > Thank you for your reply. To calculate the structure similarity, we only train the model once. Thus, there is no issue related to initialization since there is only one training process. The way we calculate the structure similarity is that we use the final model as the reference model, and use the models saved at each training epoch to measure the structural similarity with the reference model. Theoretically, the structural similarity will always go to 100%.
> > >
> > > **The initialization does not affect the analysis of structure similarity at all.**
> > >
> > >
> > > We are not clear why the reviewer mentioned it needs a lot of work to be reproducible, as we have validated the structure similarity for both MEST and RigL (via multiple runs).
> > >
> > > We hope the explanation is clearer now.

---

> ### Author Response · Authors · 2022-08-02
> **Author Response to Reviewer AURL (Part 2/6)**
>
> **Q3. Only the result on CIFAR100 using ResNet32 in the main paper. Inconsistency in achieving better performance (Clarification on ImageNet and more models with consistently better performance).**
>
> In the main paper, we have shown the results of the ResNet50 on ImageNet. Please refer to Table 3 and Section 5.1 (Line 328-333).
>
> We would like to kindly clarify that all the results in the main paper and Appendix demonstrate that our proposed method achieves consistently better performance than the reference works.
> Compared to the reference works, our method either achieves higher accuracy with similar training FLOPs, or similar accuracy with lower training FLOPs.

---

> > ### Comment · Reviewer_AURL · 2022-08-09
> > **Rebuttal Response**
> >
> > (3) Again, given your stated motivation is to improve training FLOPS, when I look at Table A6/A7 I see that training flops have actually increased significantly over most of the compared methods/results, e.g. RiGL is 0.32 training flops compared to SpFDE which is 0.36-0.55 depending on the method. We must also consider that SpFDE is using data sieving and the RiGL method is not, so this is not a fair comparison to begin with. What would the RiGL generalization result be with data sieving? I wouldn't be surprised if it was similar and reduced FLOPS.

---

> > > ### Author Response · Authors · 2022-08-09
> > > **Author response**
> > >
> > > First, we assume the reviewer is aware that we have results on ImageNet with ResNet50 in the main paper. The claim from the reviewer that only results on “a single dataset (CIFAR-100)” is not appropriate.
> > >
> > > Second, we humbly think the reviewer might have some misunderstanding of our work and the comparison. The proposed techniques, layer freezing and data sieving, can be combined with existing dynamic sparse training algorithms without modifying them. In our work, we mainly adopt MEST since it is the state-of-the-art dynamic sparse training algorithm. We can effectively reduce the FLOPs of the original MEST from 1.17 to 0.84 while maintaining a **similar and even a bit higher performance** on ImageNet, as shown in Table 3.
> > >
> > > As for the comparison with RigL, the suggested comparison from the reviewer is not appropriate. We would like to mention that a more **reasonable comparison** with RigL should be conducted in two ways:  accuracy comparison under the same training FLOPs reduction or FLOPs reduction comparison under the same accuracy. Our method actually **achieves better results than RigL** under the same FLOPs reduction. For example, in Table 3, with the same training FLOPs, i.e., 0.74, our method has higher accuracy than RigL, 75.35 vs. 74.6, on ImageNet for ResNet 50.
> > >
> > > We humbly think our comparison with RigL is accurate and fair. The reviewer suggested adding layer freezing and data sieving to RigL. Adding them to RigL can **indeed further reduce the FLOPs without performance drop**, as can be seen in Table R, Q7 of our response. Improving RigL with our proposed technique only proves our method is robust and can be applied to more dynamic sparse training algorithms.

---

> ### Author Response · Authors · 2022-08-02
> **Author Response to Reviewer AURL (Part 1/6)**
>
> **We appreciate the valuable comments from the reviewer. We will modify the suggested claim, discuss the relevant FreezeNet, and correct the typos. All other questions are addressed as follows. We hope our response can help alleviate the reviewer's concern.**
>
> ---
>
> **Q1. Clarification on FLOPs calculation.**
>
> We thank the reviewer for the question.
>
> First, we would like to kindly clarify that all our FLOPs results in the paper have already counted the FLOPs of the forward propagation of the frozen layers.
> We have mentioned this in our paper, as in lines 252~255:
> “We consider the frozen blocks still need to conduct forward propagation during training. Therefore, we compute the training FLOPs reduction of freezing a block as its sparse back-propagation computation FLOPs multiplied by the total frozen epochs of the block.”
>
> Therefore, we humbly think **our calculations of theoretical FLOPs are correct.** The FLOPs reductions of our layer freezing technique are attributed to avoiding the backpropagation of the frozen layers instead of the forward propagation.
>
> Second, we show the details of how we compute the overall training FLOPs.
>
> - **FP_i**: Forward-prop FLOPs of layer_i for one training example in dense (without considering sparsity)
> - **BP_i**: Backward-prop FLOPs of layer_i for one training example in dense (without considering sparsity)
> - **Sp_i**: Average sparsity of layer_i throughout the entire training
> - **TrainSet_size**: number of training examples remained in the training dataset (after removing partial training examples)
> - **Ep_total**: total training epochs
> - **Ep_active_i**: number of epochs for actively training (i.e., non-frozen) of layer_i
> - **FLOPs_i**: total training FLOPs of layer_i
> - **FLOPs_total**: total training FLOPs of the model
>
> Training FLOPs computation:
> ~~~~
> FLOPs_i = TrainSet_size * (1 - Sp_i) * ((Fp_i * Ep_total) + (Bp_i * Ep_active_i))
> FLOPs_all = sum(FLOPs_i),  for i = (1, 2, 3, … N)
> ~~~~
>
> We can see that:
>
> 1. The overall training FLOPs of the model are the summation of each layer’s training FLOPs.
>
> 2. For each layer’s training FLOPs, FLOPs consist of forward propagation and backward propagation. The forward propagation is counted throughout the entire training process and will not be affected by freezing. The backpropagation is only counted in the active training stage and is not counted in the frozen stage.
>
> 3. The dataset and sparsity can proportionally reduce the training FLOPs for both forward and backward propagation.
>
> Thanks for the suggestions, and we will modify the draft to further clarify the details of the FLOPs calculation.
>
> ---
>
> **Q2. Gap between the FLOPs reduction and real-world acceleration.**
>
> We totally agree with your opinion that the high acceleration from unstructured sparsity requires specific compilers and might not be able to be achieved on any hardware. We are on the same side.
>
> Actually, we are very careful about using the word “acceleration” in our paper.
> For our methods and reference works, because of the existence of unstructured sparsity, we only used the words “reduce/save training costs” instead of using the word “acceleration” throughout the entire paper, even though our methods can indeed achieve actual real-world acceleration. This is because we take this problem as **seriously** as you and hope our paper can be **rigorous**.
>
> We have also discussed this problem in our paper in Section 5.3 (Line 354-355) and Appendix E (Line 683-702), such as “Currently, the actual acceleration by leveraging weight sparsity is still limited even at a very high sparsity ratio.”
>
> How to accelerate the sparsity training is also one of the motivations of our paper. We are seeking for other possible directions to improve sparse training other than keep increasing the unstructured sparsity as the most current sparse training works did.
>
> The reason that we use FLOPs to measure possible training cost saving is to have fair comparisons with prior works. Besides FLOPs, we indeed measured the real-world acceleration achieved by our layer freezing and data sieving method.
>
> The results and discussions are shown in Table A8 and Table A9 in Appendix E. For example, layer freezing with 25% FLOPs reduction can achieve 23.67% speed up, and data sieving with 25% FLOPs reduction can achieve 24.39% speed up. Both our layer freezing and data sieving methods achieve nearly linear real-world acceleration according to the FLOPs reduction.
> Most importantly, as we mentioned on Line 700 in Appendix, “the layer freezing and data sieving methods have a high degree of practicality since the acceleration can be easily achieved using native PyTorch/TensorFlow without additional support.”
>
> The advantages of real-world acceleration make our methods valuable for the sparse training to achieve actual speedup. Our work explores two promising directions, and we hope our work can motivate more efforts towards training acceleration on real hardware for the sparse training society.

---

> > ### Comment · Reviewer_AURL · 2022-08-09
> > **Rebuttal Response**
> >
> > I thank the authors for their rebuttal and attempts to answer questions I asked in the review. I've responded to each rebuttal point/question below (continued over the multiple posts).
> >
> > (1) Thanks for the clarification that FLOPS calculation includes the forward pass, sorry if I missed this in the text. While the calculations of FLOPS might be correct, you have to provide the details of how they are computed explicitly in the paper as there are many potential ways of calculating FLOPS, so I'm happy you will include these details.
> >
> > (2) Happy the authors understand/accept the issues with theoretical FLOPS, but what do the authors mean by "reduce/save training costs"  specifically if it doesn't mean that the method is amenable to acceleration or presents any method of acceleration? Even in the abstract the motivation is listed as:
> >
> > "This paper intends to explore other possible directions to effectively and efficiently reduce sparse training costs while preserving accuracy."
> >
> > If the motivation is different, this paper will need significant work to make that motivation clear, because at the best the motivation is very misleading.
> >
> > I did miss the small number of results in Table A8/9 in the appendix, and appreciate the pointer to these. Obviously these are not a substantive evaluation, but I do believe these should be referred to in the main text to provide the reader with some real-world context.

---

> > > ### Author Response · Authors · 2022-08-09
> > > **Author response**
> > >
> > > 1). We appreciate the reviewer acknowledging did not read our explanation on how to calculate FLOPs in our paper during the review process.
> > >
> > > We indeed include a detailed description of the calculation of FLOPs in the main paper, such as Lines 252~255 and Algorithm 1. We believe that the way we calculate FLOPs for sparse training is accurate by following existing works. Additionally, to the best of our knowledge, there is only one standard way to calculate the correct FLOPs.
> > >
> > > ---
> > >
> > > 2). We would like to re-emphasize the goal of our paper. We intend to explore more directions to further reduce the training costs of sparse training while maintaining accuracy.
> > >
> > > We humbly think the reviewer might confuse the motivation of our paper. The reviewer thinks our paper works towards the training acceleration of dynamic sparse training algorithms, which is **not correct**. Our paper does not intend to modify the dynamic sparse training algorithm, such as MEST and RigL, to accelerate them. Instead, we explore other directions that can help **reduce training costs** and can be used with dynamic training algorithms. We think our motivation is clear and accurate.
> > >
> > > In fact, the goal of sparse training is not only about training acceleration. It also includes energy and memory saving, etc. In other words, training cost savings are Not Equal To training acceleration. To indicate various aspects of sparse training performance in general, FLOPs is still the **best-suited** metric. Moreover, the majority of the sparse training works use FLOPs as the metric for a fair comparison. Therefore, it is appropriate to use FLOPs in our paper to evaluate training cost saving.
> > >
> > > **We would like to emphasize again, our work focuses on training cost saving, but our work can indeed achieve real-world acceleration and memory savings at the same time.**

---

> ### Author Response · Authors · 2022-08-05
> **Author Response to Reviewer AURL**
>
> Dear Reviewer AURL,
>
> We appreciate your time and reviewing efforts to help improve our work. Thanks!
>
> We follow your initial suggestions to provide additional results, such as more ablation studies for the performance of the proposed techniques (Q6) and the performance of using RigL (Q4 & Q7).
>
> We also answer your other major questions, such as the way to calculate computation FLOPs (Q1), the gap between FLOPs reduction and real-world acceleration (Q2), and the observation of our experiment for structural similarity seeming to contradict some well-known issues (Q4).
> We hope our response to these questions could help alleviate your concerns and further demonstrate the advantage of our work.
>
> As the deadline for the author-reviewer discussion is approaching, we would sincerely appreciate it if you could kindly let us know whether our response addressed your concerns, and please let us know if you have further questions. It will be our great pleasure if you would consider updating your review or score.
>
> Best,
>
> Authors

---

> ### Author Response · Authors · 2022-08-09
> **Looking forward to your feedback**
>
> Dear Reviewer AURL,
>
> We would like to sincerely thank you again for your thoughtful suggestions and valuable feedback to improve our work.
>
> We provide additional explanations to help clarify our work. As the deadline for open discussion is soon, we would sincerely hope to use this opportunity to see if our responses are sufficient and if any concern remains. It will be our great pleasure if you would consider updating your review or score.
>
> Thanks again for your time.
>
> Best,
>
> Authors

---

### Official Review · Reviewer_363A · 2022-07-11

**Rating:** 7
**Confidence:** 4
**Soundness:** 4 excellent
**Presentation:** 4 excellent
**Contribution:** 3 good

**Summary:**

This paper proposes a framework for efficient sparse training. Instead of focusing on pushing the sparsity levels, the authors investigate the effect of two known methods, including layer freezing and data sieving, in the dynamic sparse training framework. The authors show why layer freezing can be applied to sparse training. Then, they described a framework that reduces both memory requirement and FLOPs count while achieving a comparable performance or outperforming state-of-the-art sparse training algorithms. The authors performed an extensive evaluation to support their claims.

**Questions:**

- I suggest the authors add a brief explanation about the CKA metric and how it is calculated in the background information.
- If enough time and resources are available, it would be nice to see how much the layer freezing and data sieving can be increased without a significant drop in accuracy. E.g., to present in a 2d-plot where the x-axis shows the change in one of these components and the y-axis shows the accuracy.
-  Minor: typosà lines 288 (,), 305 (sparse)

**Limitations:**

Yes, the authors have discussed the limitations and adequately addressed them.

**Strengths And Weaknesses:**

Strength:

- The paper is very well-written and has decent organization, and it was a pleasant read.
-  This paper provides insights on other approaches to push further the efficiency of sparse training algorithms instead of only focusing on increasing the sparsity level.
- Extensive evaluation that supports the initial claims.
-  The results on the ImageNet dataset are significant in decreasing the training FLOPs while increasing the accuracy and having different trade-offs between efficiency and accuracy.

Weakness:

- The effect of each component (layer freezing and data sieving) on the accuracy is not separately clear. It would be nice to add an ablation to analyze the effect of each component.
- It is unclear how the authors determine the value of T_{frz} (starting freezing). Is it difficult to choose a value for this hyperparameter for different sparsity levels/models/datasets?

---

> ### Author Response · Authors · 2022-08-02
> **Author Response to Reviewer 363A**
>
> **We thank the reviewer for the positive feedback and thoughtful comments. We are glad to know the reviewer has a pleasant read on the paper, and agrees our work proposes new approaches for pushing the efficiency for sparse training, provides extensive evaluation to support our claims, and shows significant results on ImageNet for decreasing the training FLOPs while increasing the accuracy. We will correct the suggested typo and add background information about CKA. Other questions are answered as follows.**
>
> ---
>
> **Q1. How to determine the value of $T_{frz}$.**
>
> Thank you for the question. Our $T_{frz}$ is actually calculated according to the target training FLOPs reduction ratio (e.g., 10%, 15%, 20%). In specific, we empirically choose to freeze 2/3 layers of the model gradually during the training process. At the $T_{frz}$ epoch, for every T epochs (i.e., 5 epochs, keeping the same as the DST structure changing epochs), we freeze the layers in the next block (i.e., the residual block in ResNet). We freeze the layers sequentially from the first layer/block. The per epoch training FLOPs of a layer are determined by the layer’s type and size. Therefore, given the overall target training FLOPs, the $T_{frz}$ can be easily calculated in advance.
> We use this methodology for different networks, datasets, and target FLOPs reduction ratios.
> We will clarify this in our final version.
>
> ---
>
> **Q2: Ablation for layer freezing and data sieving.**
>
> Thank you for the very constructive suggestion! Here we provide the ablation analysis for the impact of layer-freezing and data sieving on accuracy by themselves (Table F and Table G). The results are obtained using ResNet32 on the CIFAR100 with the sparsity of 60% and 90%. The accuracy results are the average value of 3 runs using random seeds.
>
> **Table F. Ablation analysis on different layer freezing ratios**
> >|  FLOPs reduction | None  | 10%   | 15%   | 20%   | 25%   | 27.5% | 30%   | 32.5% | 35%   |
> |--------------|-------|-------|-------|-------|-------|-------|-------|-------|-------|
> | sparsity 60% | 73.97 | 74.05 | 74.09 | 73.76 | 73.27 | 73.14 | 73.03 | 72.36 | 72.00 |
> | sparsity 90% | 71.30 | 71.33 | 71.31 | 71.29 | 71.18 | 71.08 | 70.82 | 70.35 | 70.26 |
>
> **Table G. Ablation analysis on different data sieving ratios**
> >| FLOPs reduction | None  | 10%   | 15%   | 20%   | 25%   | 27.5% | 30%   | 32.5% | 35%   |
> |--------------|-------|-------|-------|-------|-------|-------|-------|-------|-------|
> | sparsity 60% | 73.97 | 73.98 | 73.94 | 73.88 | 73.66 | 73.68 | 73.58 | 73.55 | 73.20 |
> | sparsity 90% |  71.3 | 71.36 | 71.30 | 71.33 | 71.11 | 71.09 | 70.98 | 70.86 | 70.59 |
>
>
> From the experiments, we can further find some interesting observations:
> 1. Under both sparsity of 60% and 90%, saving up to 15% training costs (FLOPs) via either layer freezing or data sieving does not lead to any accuracy drop.
>
> 2. When under a higher sparsity ratio (90% vs. 60%), sparse training can tolerate a higher FLOPs reduction for both layer freezing and data sieving. For example, compared to the non-freezing case (i.e., None in the second column), the layer freezing with a 20% FLOPs reduction leads to a -0.01% and -0.21% accuracy drop for 90% sparsity and 60% sparsity, respectively. As for the data sieving, compared to the non-freezing case, under a 20% FLOPs reduction, there is a -0.19% and -0.31% accuracy drop for 90% sparsity and 60% sparsity, respectively. We think the reason is, under a higher sparsity ratio, the upper bound for model accuracy/generalization capability is decreased, mitigating the sensitivity to the number of training data or layer freezing.
>
> 3. With a relatively higher FLOPs reduction ratio (i.e., 30%~35%), data sieving preserves higher accuracy than layer freezing under the same FLOPs reduction ratio. This inspires that if people intend to pursue a more aggressive FLOPs reduction at the cost of accuracy degradation, removing more data via the data sieving method is a more desirable choice than freezing more layers.
>
>
> We further show a comparison between only using layer-freezing or data sieving, or both of them to achieve similar FLOPs reductions, in the following table (Table H).
>
> **Table H. Analysis of layer freeze, data sieving, or both of them for FLOPs reduction.**
> >|  | freeze + data sieve | freeze only | data sieve only |
> |---:|---:|---:|---:|
> | FLOPs reduction | 27.75% (15%+15%) | 27.50% | 27.50% |
> | Accuracy  | 71.35 | 71.08 | 71.09 |
> | FLOPs reduction | 36% (20%+20%) | 35.00% | 35.00% |
> | Accuracy  | 71.25 | 70.26 | 70.59 |
>
>
> It can be observed that to achieve similar FLOPs reduction, using layer-freezing and data sieving together achieves much higher accuracy than by only using one of them individually, showing the importance of combining the two techniques.
>
> We will definitely include these ablation studies and these interesting observations in our final version. We hope our work can motivate more future research in the sparse training area.

---

> ### Author Response · Authors · 2022-08-09
> **Author Response to Reviewer 363A**
>
> Dear Reviewer 363A,
>
> **Thanks for your time and reviewing efforts to help improve our work! We appreciate your positive rating and thoughtful comments.**
>
> We provide clarification of our method and the results of the ablation study as suggested. We hope our responses have answered your questions.
>
> Best,
>
> Authors

---

> ### Comment · Reviewer_363A · 2022-08-09
> **Response**
>
> I thank the authors for the clarifications and for performing the ablation study to show the effect of layer freezing and data sieving on accuracy. My concerns are addressed, and I maintain the same score.

---

> > ### Author Response · Authors · 2022-08-09
> > **Thanks for the positive feedback**
> >
> > Dear Reviewer 363A,
> >
> > We would like to thank you again for your time and your positive rating. This is a great affirmation of our work.
> >
> > Best,
> >
> > Authors

---

### Official Review · Reviewer_EhKj · 2022-07-11

**Rating:** 6
**Confidence:** 4
**Soundness:** 3 good
**Presentation:** 3 good
**Contribution:** 3 good

**Summary:**

This paper brings a new perspective in sparse neural networks training with dynamic sparsity (or, on short, DST) by studying layer freezing and data sieving. Moreover, it highlights and brings understanding on why DST is fundamentally different than dense training. Consequently, it proposes a new framework to improve DST performance by considering three dimensions, i.e., weight Sparsity, layer Freezing, and Dataset Efficiency (SpFDE). The empirical validation shows good performance for SpFDE.

**Questions:**

Structural similarity. Line 163 – 50% seems an arbitrary choice? How other values would affect the structural similarity findings?

Representational similarity. Line 188 – Why you find this “surprisingly”? As long as connections are added and removed from the networks in DST during learning, the representation is somehow “forced” to change.

Training set. Lines 288-289 – Is there any relation with bagging?


**Strengths And Weaknesses:**

**Strengths**

_ Originality. Up to my best knowledge, this paper is the first to study layer Freezing in Sparse Training. The study itself is novel, and the findings on structural and representational similarity bring a new perspective in DST.

_ Quality. The novel proposed methodology and the well-designed empirical validation seem sound. Good qualitative discussions, reflecting a good understanding of DST and its challenges.

_ Clarity. The paper is well written.

_ Significance. The paper may have some impact. Particularly, the knowledge unveiled on structural and representational similarity can lay the ground for impactful future research.

**Weaknesses**

_ Clarity. A careful proofread is recommended. There are still typos (e.g., on line 305)

_ Statements. Some statements are too strong. Please consider revising. For instance, line 326, *reducing training costs by pushing sparsity towards extreme ratios is no longer a desirable methodology* prunes valid research directions.

_ (not really a weak point) The improvement in performance over the well-chosen baselines is relatively small, but quite constant.

---

> ### Author Response · Authors · 2022-08-02
> **Author Response to Reviewer EhKj (Part 2/2)**
>
> **Q3. About representation similarity. Why do you find this “surprisingly”?**
>
> We would intuitively think that the representation learning speed of a layer might be significantly affected by its sparsity ratio since the sparsity could affect the model’s capacity and generalizability. The dynamically changed sparse structure may also affect the representation learning speed. However, our experiential results reveal that the representation learning speed is not affected by the model sparsity ratio. Though the upper bound of the model/layer’s capacity might be decreased, the convergence speed to the upper bound is not affected. Due to the fact that only the least significant weights are altered during the dynamically changed sparse structure, it only has a minor effect on the representation learning speed.
>
> This interesting observation is somehow counterintuitive, and that is the reason we find it “surprisingly”.
>
> The representation learning speed is critical to our layer freezing technique. The results indicate that the layer freezing technique can potentially be adopted in the sparse training process as early as in the dense model training process, thereby effectively reducing the training costs.
>
> Additionally, we evaluate the representation learning speed using another representative dynamic sparse training method, i.e., RigL, instead of MEST. Similar observations can be obtained from the following table (Table E). That is, the sparsity ratio will not lead to an apparent change in the network representation learning speed.
>
> >**Table E. Trend of representational similarity: the same layer (10th) with different sparsity. RigL is used for dynamic sparse training.**
> | sparsity \ epoch | 1 | 20 | 40 | 60 | 80 | 100 | 120 | 140 | 160 |
> |---|---:|---:|---:|---:|---:|---:|---:|---:|---:|
> | Dense | 0.150 | 0.903 | 0.958 | 0.969 | 0.968 | 0.981 | 0.998 | 0.999 | 1.000 |
> | sp0.5 | 0.200 | 0.834 | 0.941 | 0.958 | 0.969 | 0.990 | 0.992 | 0.999 | 1.000 |
> | sp0.8 | 0.151 | 0.899 | 0.924 | 0.972 | 0.988 | 0.988 | 0.994 | 0.997 | 1.000 |
> | sp0.9 | 0.251 | 0.908 | 0.966 | 0.971 | 0.977 | 0.981 | 0.992 | 0.999 | 1.000 |
>
> ---
>
> **Q4: Relation between our training set and bagging.**
>
> Thanks for the comment. There are some similarities between our adopted strategy for training set preparation and bagging. For example, the training set for each epoch is a subset of the whole training data. However, since we study the training of a single sparse network, we do not apply the ensemble of multiple models used in bagging for performance improvement.

---

> ### Author Response · Authors · 2022-08-02
> **Author Response to Reviewer EhKj (Part 1/2)**
>
> **We would like to thank the reviewer for the positive feedback and valuable review. We appreciate the reviewer's acknowledgment that our novel study on layer freeze in sparse training brings a new perspective to dynamic sparsity training (DST); the proposed methodology is novel; validation is well designed and sound; good qualitative discussions reflect a good understanding of DST; and the paper is well written and impactful for future research. We will correct the typo, and other questions are addressed as follows.**
>
> ---
>
> **Q1: About statement revising.**
>
> Thanks for the kind suggestion. We will revise the statement as “pushing sparsity towards extreme ratios is not the only direction for reducing training costs.”
>
> ---
>
> **Q2: About structure similarity.**
>
> For the structural similarity, the reason that we use 50% most significant non-zero weights is that the sparse training algorithm, i.e., MEST, forces the 50% less important weights/mask to be changed, and the most significant weights play the most important role in the model’s generalizability. Thus, tracking the structural similarity using 50% most significant non-zero weights is reasonable.
>
> Here we provide the analysis of structural similarity traces measured using 70% most significant non-zero weights and all (100%) non-zero weights in Table A and Table B. All the results are based on 90% weight sparsity.
>
>
> >**Table A. Structure similarity (%) using 70% most significant non-zero weights (with MEST)**
> | layer \ epoch |    1 |   20 |   40 |   60 |   80 |   100 |   120 |   140 |   160 |
> |---------------|-----:|-----:|-----:|-----:|-----:|------:|------:|------:|------:|
> | layer 1       | 37.5 | 73.4 | 89.2 | 93.8 | 97.5 | 100.0 | 100.0 | 100.0 | 100.0 |
> | layer 10      | 20.8 | 48.2 | 67.6 | 84.7 | 87.6 |  93.5 | 100.0 | 100.0 | 100.0 |
> | layer 17      | 19.9 | 46.2 | 63.4 | 78.3 | 83.7 |  91.8 |  98.2 | 100.0 | 100.0 |
> | layer 25      | 15.3 | 42.5 | 51.5 | 63.0 | 71.0 |  82.7 |  95.6 | 100.0 | 100.0 |
>
>
> >**Table B. Structure similarity (%) using 100% non-zero weights (with MEST)**
> | layer \ epoch |    1 |   20 |   40 |   60 |   80 |  100 |   120 |   140 |   160 |
> |---------------|-----:|-----:|-----:|-----:|-----:|-----:|------:|------:|------:|
> | layer 1       | 30.2 | 65.1 | 76.7 | 82.6 | 91.3 | 94.8 | 100.0 | 100.0 | 100.0 |
> | layer 10      | 19.5 | 40.2 | 56.7 | 73.6 | 85.7 | 89.3 |  94.5 | 100.0 | 100.0 |
> | layer 17      | 19.7 | 39.1 | 52.7 | 66.0 | 78.6 | 82.1 |  90.3 | 100.0 | 100.0 |
> | layer 25      | 13.6 | 37.4 | 46.5 | 54.1 | 63.6 | 73.1 |  81.3 |  89.2 | 100.0 |
>
> It can also be observed from the above two tables that the structural similarity of the earlier layers converges significantly faster than the later layers. Therefore, the conclusion from using 70% and 100% most significant non-zero weights is consistent with using 50% of the most significant non-zero weights.
>
> Additionally, we evaluate the structural similarity using RigL instead of MEST, and the structural similarity convergence situation can be observed in the following tables (Table C and Table D):
>
> >**Table C. Structure similarity (%) using 50% non-zero weights (with RigL)**
> | layer \ epoch |    1 |   20 |   40 |    60 |    80 |   100 |   120 |   140 |   160 |
> |---------------|-----:|-----:|-----:|------:|------:|------:|------:|------:|------:|
> | layer 1       | 39.5 | 93.0 | 97.7 | 100.0 | 100.0 | 100.0 | 100.0 | 100.0 | 100.0 |
> | layer 10      | 37.7 | 77.8 | 89.5 |  94.4 |  99.3 | 100.0 | 100.0 | 100.0 | 100.0 |
> | layer 17      | 20.7 | 65.8 | 76.2 |  90.3 |  97.4 | 100.0 | 100.0 | 100.0 | 100.0 |
> | layer 25      | 19.1 | 60.4 | 65.1 |  89.3 |  92.1 | 100.0 | 100.0 | 100.0 | 100.0 |
>
> >**Table D. Structure similarity (%) using 70% non-zero weights (with RigL)**
> | layer \ epoch |    1 |   20 |   40 |   60 |    80 |   100 |   120 |   140 |   160 |
> |---------------|-----:|-----:|-----:|-----:|------:|------:|------:|------:|------:|
> | layer 1       | 31.7 | 86.7 | 93.3 | 98.3 | 100.0 | 100.0 | 100.0 | 100.0 | 100.0 |
> | layer 10      | 31.8 | 63.5 | 73.5 | 84.3 |  92.5 |  98.8 | 100.0 | 100.0 | 100.0 |
> | layer 17      | 20.2 | 50.6 | 63.4 | 79.5 |  87.1 |  97.9 | 100.0 | 100.0 | 100.0 |
> | layer 25      | 17.7 | 49.1 | 64.0 | 74.2 |  85.4 |  97.5 | 100.0 | 100.0 | 100.0 |
>
> We can find the trends for structure similarity convergence when using RigL as the sparse training algorithm is similar to using the MEST sparse training algorithm.

---

> ### Author Response · Authors · 2022-08-09
> **Author Response to Reviewer EhKj**
>
> Dear Reviewer EhKj,
>
> **Thanks for your time and reviewing efforts to help improve our work! We appreciate your positive rating and thoughtful comments.**
>
> We provide suggested results in the authors' response, such as the clarification on structure and representation similarity. We hope our responses have answered your questions. It will be our great pleasure if you would consider updating your review or score.
>
>
> Best,
>
> Authors

---

> > ### Comment · Reviewer_EhKj · 2022-08-09
> > **Thanks for the response**
> >
> > Dear authors,
> >
> > I appreciate the fact that you considered my comments. Thank you. I find the new results in line with my expectations. I don't feel comfortable in increasing my score, but I would be happy to see this paper accepted.
> >
> > Best,
> > Reviewer EhKj

---

> > > ### Author Response · Authors · 2022-08-09
> > > **Thanks for the positive feedback**
> > >
> > > Dear Reviewer EhKj,
> > >
> > > We would like to thank you again for your time and your positive rating. We hope that our work can bring these new perspectives to the community as well.
> > >
> > > Best,
> > >
> > > Authors

---

### Official Review · Reviewer_y6sL · 2022-07-12

**Rating:** 6
**Confidence:** 3
**Soundness:** 2 fair
**Presentation:** 4 excellent
**Contribution:** 2 fair

**Summary:**

The paper studies two concepts, namely, layer freezing and data sieving, in conjunction with sparse training for DNN architectures. The layer freezing methodology is not new, but it has been studied for the first time in this context. The newly proposed data sieving methodology relies on first removing a % of the training instances and then at each epoch, a similar % of the easiest instances are removed and replaced with instances removed from the original set. Both ideas are studied in the context of sparse training, and extensive experimentation shows promising improvement in reducing computational and memory costs with good accuracy.

**Questions:**

I enjoyed reading the paper as it's well-written and the two concepts studied are properly introduced and motivated by experiments to demonstrate their flaws and importance.

The experimental results support the claims, and they are extensive.

The code is provided, which can help in the validation of the results as well as in the adaptation of the proposed solution.

My main concern is that the paper is lacking in terms of technical depth. The ideas explored are relatively simple, and the idea of sieving reminds me of ideas of perturbation analysis (we don't perturb but instead remove instances and replace the easily classified cases). That being said, sometimes we don't know over-complicated solutions, and this approach seems to work well.



**Strengths And Weaknesses:**

Strengths:

S1. Important and timely problem
S2. Well-written paper, nicely motivated
S3. Extensive experimental results

Weaknesses:

W1. The technical depth is somewhat low

---

> ### Author Response · Authors · 2022-08-02
> **Author Response to Reviewer y6sL**
>
> **We appreciate the reviewer for acknowledging our work studies an important and timely problem, proposes nicely motivated ideas, and shows extensive experimental results to support the proposed concepts. We are glad to know the reviewer enjoys reading our paper.**
>
> We agree with the reviewer that layer freezing, which is first studied under the context of dynamic sparse training, and data sieving are direct techniques to improve sparse training by reducing computational and memory costs.
> We also would like to kindly mention that, besides the proposed layer-freezing and data-sieving methods, **our analysis of structural and representational similarity** in dynamic sparse training (DST) are also the far-reaching contributions that unveil certain inherent properties of DST, and this could be **insightful for further research**.
>
> The simplicity of the proposed methods is also attributed to our **extensive explorations** such as the impact of the different schemes and criteria. We hope those results and analyses could further provide **valuable insights and experience** for further work.
>
>
> On the other hand, the simplicity of the two approaches is especially **beneficial for sparse training** since their implementation neither introduces extra computation overhead nor requires support from specific compilers. We show in Appendix Table A8 and Table A9 that both methods achieve almost the linear training acceleration according to the FLOPs reduction, demonstrating negligible computation overhead. For example, layer freezing with 25% FLOPs reduction can achieve a 23.67% speed up, and data sieving with 25% FLOPs reduction can achieve a 24.39% speed up.
> As for the data sieving, we agree with the reviewer that we remove the easier classifier samples instead of performing perturbing. Thanks for the suggestion, and we will add more discussion regarding the implementation of layer freezing and data sieving and their advantages for sparse training.

---

> ### Author Response · Authors · 2022-08-09
> **Author Response to Reviewer y6sL**
>
> Dear Reviewer y6sL,
>
> **We would like to sincerely thank you again for your valuable feedback. It is our great pleasure to know you enjoyed reading our paper.**
>
> We provide further clarification about the design philosophy of our techniques. We hope our responses have answered your questions.
>
> Best,
>
> Authors

---

### Meta-Review · Area_Chair_MWZf · 2022-08-21

**Recommendation:** Accept
**Confidence:** Less certain

**Metareview:**

The paper provides methods to improve the task of sparse training. The reviews agree that the idea is well motivated, novel and that the paper brings insights to sparse training that would be of interest to the community. The experiments seem quite extensive and show that these methods allow to improve the Pareto curve of training process FLOPs vs obtained model quality. One of the reviews raised several issues about the paper, questioning the soundness of the experiments and method. After reviewing the discussion, the major issues seem to be not fundamental flaws but unclear details in the paper. Since these are clarified in the discussion, I view these as minor issues that can be fixed towards a camera ready version, and I urge the authors to carefully go over the reviews and fix the paper to be more clear.
Concluding, the consensus around novelty and overall positive feedback that remained positive through the discussion phase, lead me to believe the advantages of the paper outweigh its flaws.


**Award:**

No

---

### Decision · Program_Chairs · 2022-09-14

Accept